

**Development and evaluation of a variably saturated flow model in the global**
**E3SM Land Model (ELM) Version 1.0**
**Gautam Bisht[1], William J. Riley[1], Glenn E. Hammond[2], and David M. Lorenzetti[3]**
[1]Climate & Ecosystem Sciences Division, Lawrence Berkeley National Laboratory, 1
Cyclotron Road, Berkeley, California 94720, USA
[2]Applied Systems Analysis and Research Department, Sandia National Laboratories,
Albuquerque, NM 87185-0747, USA
[3]Sustainable Energy Systems Group, Lawrence Berkeley National Laboratory, 1
Cyclotron Road, Berkeley, California 94720, USA
Correspondence to: Gautam Bisht (gbisht@lbl.gov)





**Abstract**
Improving global-scale model representations of coupled surface and groundwater
hydrology is important for accurately simulating terrestrial processes and predicting
climate change effects on water resources. Most existing land surface models,
including the default E3SM Land Model (ELMv0), which we modify here, routinely
employ different formulations for water transport in the vadose and pheratic zones.
In this work, we developed the Variably Saturated Flow Model (VSFM) in ELMv1 to
unify the treatment of soil hydrologic processes in the unsaturated and saturated
zones. VSFM was tested on three benchmark problems and results were evaluated
against observations and an existing benchmark model (PFLOTRAN). The ELMv1-
VSFM's subsurface drainage parameter, $f_d$ , was calibrated to match an
observationally-constrained and spatially-explicit global water table depth (WTD)
product. An optimal $f_d$ was obtained for 79% of global $1.9^0 \times 2.5^0$ gridcells, while the
remaining 21% of global gridcells had predicted WTD deeper than the
observationally-constrained estimate. Comparison with predictions using the default
$f_d$ value demonstrated that calibration significantly improved prediction, primarily
by allowing much deeper WTDs. Model evaluation using the International Land Model
Benchmarking package (ILAMB) showed that improvements in WTD predictions did
not degrade model skill for any other metrics. We evaluated the computational
performance of the VSFM model and found that the model is about 30% more
expensive than the default ELMv0 with an optimal processor layout.

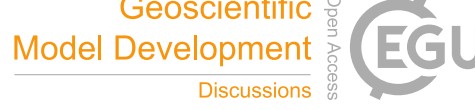



## 1   Introduction

Groundwater, which accounts for 30% of freshwater reserves globally, is a vital human water resource. It is estimated that groundwater provides 20-30% of global freshwater withdrawals (Petra, 2009; Zektser and Evertt, 2004), and that irrigation accounts for ∼70% of these withdrawals (Siebert et al., 2010). Climate change is expected to impact the quality and quantity of groundwater in the future (Alley, 2001). As temporal variability of precipitation and surface water increases in the future due to climate change, reliance on groundwater as a source of fresh water for domestic, agriculture, and industrial use is expected to increase (Taylor et al., 2013).

Local environmental conditions modulate the impact of rainfall changes on groundwater resources. For example, high intensity precipitation in humid areas may lead to a decrease in groundwater recharge (due to higher surface runoff), while arid regions are expected to see gains in groundwater storage (as infiltrating water quickly travels deep into the ground before it can be lost to the atmosphere) (Kundzewicz and Doli, 2009). Although global climate models predict changes in precipitation over the next century (Marvel et al., 2017), few global models that participated in the recent Coupled Model Inter-comparison Project (CMIP5; Taylor et al. (2012)) were able to represent global groundwater dynamics accurately (e.g. Swenson and Lawrence (2014))

Modeling studies have also investigated impacts, at watershed to global scale, on future groundwater resources associated with land-use (LU) and land-cover (LC) change (Dams et al., 2008) and ground water pumping (Ferguson and Maxwell, 2012; Leng et al., 2015). Dams et al. (2008) predicted that LU changes would result in a small mean decrease in subsurface recharge and large spatial and temporal variability in groundwater depth for the Kleine Nete basin in Belgium. Ferguson and Maxwell (2012) concluded that groundwater-fed irrigation impacts on water exchanges with the atmosphere and groundwater resources can be comparable to those from a 2.5 °C increase in air temperature for the Little Washita basin in Oklahoma, USA. By performing global simulations of climate change scenarios using CLM4, Leng et al. (2015) concluded that the water source (i.e., surface or groundwater) used for



irrigation depletes the corresponding water source while increasing the storage of
the other water source. Recently, Leng et al. (2017) showed that irrigation method
(drip, sprinkler, or flood) has impacts on water balances and water use efficiency in
global simulations.

Groundwater models are critical for developing understanding of

groundwater systems and predicting impacts of climate change (Green et al., 2011;
Kollet and Maxwell, 2008). Kollet and Maxwell (2008) identified critical zones, i.e.,
regions within the watershed with water table depths between $1 - 5$ m, where the
influence of groundwater dynamics was largest on surface energy budgets. Numerical
studies have demonstrated impacts of groundwater dynamics on several key Earth
system processes, including soil moisture (Chen and Hu, 2004; Liang et al., 2003;
Salvucci and Entekhabi, 1995; Yeh and Eltahir, 2005), runoff generation (Levine and
Salvucci, 1999; Maxwell and Miller, 2005; Salvucci and Entekhabi, 1995; Shen et al.,
2013), surface energy budgets (Alkhaier et al., 2012; Niu et al., 2017; Rihani et al.,
2010; Soylu et al., 2011), land-atmosphere interactions (Anyah et al., 2008; Jiang et
al., 2009; Leung et al., 2011; Yuan et al., 2008), vegetation dynamics (Banks et al.,
2011; Chen et al., 2010), and soil biogeochemistry (Lohse et al., 2009; Pacific et al.,

2011).

Recognizing the importance of groundwater systems on terrestrial processes,

groundwater models of varying complexity have been implemented in land surface
models (LSMs) in recent years. Groundwater models in current LSMs can be classified
into four categories based on their governing equations. Type-1 models assume a
quasi-steady state equilibrium of the soil moisture profile above the water table
(Hilberts et al., 2005; Koster et al., 2000; Walko et al., 2000). Type-2 models use a θ-
based (where θ is the water volume content) Richards equation in the unsaturated
zone coupled with a lumped unconfined aquifer model in the saturated zone.
Examples of one-dimensional Type-2 models include Liang et al. (2003), Yeh and
Eltahir (2005), Niu et al. (2007), and Zeng and Decker (2009). Examples of quasi
three-dimensional Type-2 models are York et al. (2002); Fan et al. (2007); Miguez-
Macho et al. (2007); and Shen et al. (2013). Type-3 models include a three-

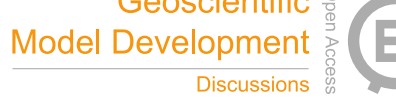



dimensional representation of subsurface flow based on the variably saturated
Richards equation (Maxwell and Miller, 2005; Tian et al., 2012). Type-3 models
employ a unified treatment of hydrologic processes in the vadose and pheratic zones
but lump changes associated with water density and unconfined aquifer porosity into
a specific storage term. The fourth class (Type-4) of subsurface flow and reactive
transport models (e.g., PFLOTRAN (Hammond and Lichtner, 2010), TOUGH2 (Pruess
et al., 1999), and STOMP (White and STOMP, 2000)) combine a water equation of
state (EoS) and soil compressibility with the variably saturated Richards equation.
Type-4 models have not been routinely coupled with LSMs to address climate change
relevant research questions.
The Energy, Exascale, Earth System Model (E3SM) is a new Earth System
Modeling project sponsored by the U.S. Department of Energy (DOE). The E3SM
model started from the Community Earth System Model (CESM) version 1_3_beta10
(Oleson, 2013). Specifically, the initial version (v0) of the E3SM Land Model (ELM)
was based off the Community Land Model's (CLM's) tag 4_5_71. ELMv0 uses a Type-
2 subsurface hydrology model based on Zeng and Decker (2009). In this work, we
developed in ELMv1 a Type-4 Variably Saturated Flow model (VSFM) to provide a
unified treatment of soil hydrologic processes within the unsaturated and saturated
zones. The VSFM formulation is based on the isothermal single phase flow model of
PFLOTRAN (see Hammond and Lichtner (2010) for details regarding various modes
supported in PFLOTRAN). While PFLOTRAN is a massively parallel, three-
dimensional subsurface model, the VSFM is a serial, one-dimensional model that is
appropriate for climate scale applications.
This paper is organized into several sections: (1) brief review of the ELMv0
subsurface hydrology model; (2) overview of the VSFM formulation integrated in
ELMv1; (3) application of the new model formulation to three benchmark problems;
(4) development of a subsurface drainage parameterization necessary to predict
global water table depths (WTDs) comparable to recently released observationally-
constrained estimates; (5) comparison of ELMv1 global simulations with the default
subsurface hydrology model and VSFM against multiple observations using the



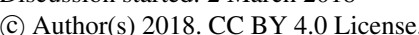


International Land Model Benchmarking package (ILAMB; Hoffman et al. (2017));
and (6) a summary of major findings.

## 2    Methods

### 2.1    Current Model Formulation

Water flow in the unsaturated zone is often described by the $\theta$-based Richards
equation:

$$\frac{\partial \theta}{\partial t} = -\boldsymbol{\nabla} \cdot \boldsymbol{q} - Q \tag{1}$$


where $\theta$ [m³ of water m⁻³ of soil] is the volumetric soil water content, $t$ [s] is time, $\boldsymbol{q}$
[m s⁻¹] is the Darcy water flux, and $Q$ [m³ of water m⁻³ of soil s⁻¹] is a soil moisture
sink term.  The Darcy flux, $\vec{q}$, is given by

$$\boldsymbol{q} = -K\boldsymbol{\nabla}(\psi + \text{z}) \tag{2}$$

where $K$ [ms⁻¹] is the hydraulic conductivity, $z$ [m] is height above some datum in the
soil column and $\psi$ [m] is the soil matric potential. The hydraulic conductivity and soil
matric potential are modeled as non-linear function of volumetric soil moisture
following Clapp and Hornberger (1978):

$$K = \Theta_{ice} K_{sat} \left(\frac{\theta}{\theta_{sat}}\right)^{2B+3} \tag{3}$$

$$\psi = \psi_{sat} \left(\frac{\theta}{\theta_{sat}}\right)^{-B} \tag{4}$$


where $K_{sat}$ [m s⁻¹] is saturated hydraulic conductivity, $\psi_{sat}$ [m] is saturated soil
matric potential, $B$ is a linear function of percentage clay and organic content (Oleson,
2013), and $\Theta_{ice}$ is the ice impedance factor (Swenson et al., 2012). ELMv0 uses the
modified form of Richards equation of Zeng and Decker (2009) that computes Darcy
flux as

$$\boldsymbol{q} = -K\boldsymbol{\nabla}(\psi + \text{z} - \text{C}) \tag{5}$$

where C is a constant hydraulic potential above the water table, $z_{\nabla}$, given as





$$C = \psi_E + \text{z} = \psi_{sat}\left(\frac{\theta_E(z)}{\theta_{sat}}\right)^{-B} = \psi_{sat} + z_\nabla \tag{6}$$

where $\psi_E$ [m] is the equilibrium soil matric potential, $\theta_E$ [m³ m⁻³] is volumetric soil
water content at equilibrium soil matric potential, and $z_\nabla$ [m] is height of water table
above the reference datum. ELMv0 uses a cell-centered finite volume spatial
discretization and backward Euler implicit time integration. By default, ELMv0's
vertical discretization of a soil column yields 15 soil layers of exponentially varying
soil thicknesses that reach a depth of 42.1 m Only the first 10 soils layers (or top 3.8
m of each soil column), are hydrologically active, while thermal processes are
resolved for all 15 soils layers. The nonlinear Darcy flux is linearized using Taylor
series expansion and the resulting tridiagonal system of equations is solved by LU
factorization.

Flow in the saturated zone is modeled as an unconfined aquifer below the soil

column based on the work of Niu et al. (2007). Exchange of water between the soil
column and unconfined aquifer depends on the location of the water table. When the
water table is below the last hydrologically active soil layer in the column, a recharge
flux from the last soil layer replenishes the unconfined aquifer. A zero-flux boundary
condition is applied to the last hydrologically active soil layer when the water table is
within the soil column. The unconfined aquifer is drained by a flux computed based
on the SIMTOP scheme of Niu et al. (2007) with modifications to account for frozen
soils (Oleson, 2013).
**2.2   New VSFM Model Formulation**
In the VSFM formulation integrated in ELMv1, we use the mass conservative form of
the variably saturated subsurface flow equation (Farthing et al., 2003; Hammond and
Lichtner, 2010; Kees and Miller, 2002):

$$\frac{\partial(\phi s_w \rho)}{\partial t} = -\boldsymbol{\nabla} \cdot (\rho \boldsymbol{q}) - Q \tag{7}$$

where $\phi$ [m³ m⁻³] is the soil porosity, $s_w$ [-] is saturation, $\rho$ [kg m⁻³] is water density,
$\boldsymbol{q}$ [m s⁻¹] is the Darcy velocity, and $Q$ [kg m⁻³ s⁻¹] is a water sink. We restrict our model





formulation to a one-dimensional system and the flow velocity is defined by Darcy's
law:

$$\boldsymbol{q} = -\frac{kk_r}{\mu}\boldsymbol{\nabla}(P + \rho gz) \tag{8}$$

where $k$ [m²] is intrinsic permeability, $k_r$ [-] is relative permeability, $\mu$ [Pa s] is
viscosity of water, $P$ [Pa] is pressure], g [m s⁻²] is the acceleration due to gravity, and
z [m] is elevation above some datum in the soil column.

In order to close the system, a constitutive relationship is used to express soil

saturation and relative permeability as a function of soil matric pressure. Analytic
Water Retention Curves (WRCs) are used to model effective saturation ($s_e$)

$$s_e = \left(\frac{s_w - s_r}{1 - s_r}\right) \tag{9}$$

where $s_w$ is soil saturation and $s_r$ is residual soil saturation. We have implemented
Brooks and Corey (1964) (equation 10) and van Genuchten (1980) (equation 11)
WRCs:

$$s_e = \begin{cases} \left(\frac{-P_c}{P_c^0}\right)^{-\lambda} & if\ P_c < 0 \\ 1 & if\ P \geq 0 \end{cases} \tag{10}$$

$$s_e = \begin{cases} [1 + (\alpha|P_c|^n)]^{-m} & if\ P < 0 \\ 1 & if\ P \geq 0 \end{cases} \tag{11}$$

where $P_c$ [Pa] is the capillary pressure and $P_c^0$ [Pa] is capillary pressure denoting air
entry point. The capillary pressure is computed as $P_c = P - P_{ref}$ where $P_{ref}$ is $P_c^0$ for
Brooks and Corey WRC and typically the atmospheric pressure (=101,325 [Pa]) is
used for van Genuchten WRC. In addition, a smooth approximation of equation (10)
and (11) was developed to facilitate convergence of the nonlinear solver (Appendix
A). Relative soil permeability was modeled using the Mualem (1976) formulation:

$$\kappa_r(s_e) = \begin{cases} s_e^{0.5}\left[1 - \left(1 - s_e^{1/m}\right)^m\right] & if\ P < P_{ref} \\ 1 & if\ P \geq P_{ref} \end{cases} \tag{12}$$

Lastly, we used an EoS for water given by Tanaka et al. (2001):

$$\rho(P,T) = [1 + (k_0 + k_1T + k_2T^2)(P - P_{ref})]a_5\left[1 - \frac{(T + a_1)^2(T + a_2)}{a_3(T + a_4)}\right] \tag{13}$$





where

$$k_0 = 50.74 \times 10^{-11} \text{ [Pa}^{-1}\text{]}$$
$$k_1 = -0.326 \times 10^{-11} \text{ [Pa}^{-1}\text{C}^{-1}\text{]}$$
$$k_2 = 0.00416 \times 10^{-11} \text{ [Pa}^{-1}\text{C}^2\text{]}$$
$$a_1 = -3.983035 \text{ [C]}$$
$$a_2 = 301.797 \text{ [C]}$$
$$a_3 = 522558.9 \text{ [C}^{-2}\text{]}$$
$$a_4 = 69.34881 \text{ [C]}$$
$$a_5 = 999.974950 \text{ [kg m}^{-3}\text{]}$$

Unlike the default subsurface hydrology model, the VSFM is applied over the

full sol depth (in the default model, 15 soils layers). The VSFM model replaces both
the $\theta$-based Richards equation and the unconfined aquifer of the default model. In the
VSFM model, water table depth is diagnosed based on the vertical soil liquid pressure
profile. Like the default model, drainage flux is computed based on the modified
SIMTOP approach and is vertically distributed over the soil layers below the water
table.

### 2.2.1   Discrete Equations

We use a cell-centered finite volume discretization to decompose the spatial

domain, $\Omega$, into N non-overlapping control volumes, $\Omega_n$, such that $\Omega = \cup_{n=1}^{N} \Omega_i$ and $\Gamma_n$
represents the boundary of the $n$-th control volume. Applying a finite volume integral
to equation (7) and the divergence theorem yields

$$\frac{\partial}{\partial t} \int_{\Omega_n} (\phi s_w \rho) \, dV = - \int_{\Gamma_n} (\rho \boldsymbol{q}) \cdot d\boldsymbol{A} - \int_{\Omega_n} Q \tag{14}$$

The discretized form of the left hand side term and first term on the right hand side
of equation (14) are approximated as:

$$\frac{\partial}{\partial t} \int_{\Omega_n} (\phi s_w \rho) \, dV \approx \left( \frac{d}{dt} (\phi s_w \rho) \right) V_n \tag{15}$$



$$\int_{\Gamma_n} (\rho\boldsymbol{q}) \cdot d\boldsymbol{A} \approx \sum_{n'} (\rho\boldsymbol{q})_{nn'} \cdot \boldsymbol{A}_{nn'} \tag{16}$$

After substituting equations (15) and (16) in equation (14), the resulting ordinary
differential equation for the variably saturated flow model is

$$\left(\frac{d}{dt}(\phi s_w \rho)\right) V_n = -\sum_{n'} (\rho\boldsymbol{q})_{nn'} \cdot \boldsymbol{A}_{nn'} - Q_n V_n \tag{17}$$

We perform temporal integration of equation (17) using the backward-Euler scheme:

$$\left(\frac{(\phi s_w \rho)_n^{t+1} - (\phi s_w \rho)_n^t}{\Delta t}\right) V_n = -\sum_{n'} (\rho\boldsymbol{q})_{nn'}^{t+1} \cdot \boldsymbol{A}_{nn'} - Q_n^{t+1} V_n \tag{18}$$

Rearranging terms of equation (18) results in a nonlinear equation for the unknown
pressure at timestep $t+1$ as

$$\left(\frac{(\phi s_w \rho)_n^{t+1} - (\phi s_w \rho)_n^t}{\Delta t}\right) V_n + \sum_{n'} (\rho\boldsymbol{q})_{nn'}^{t+1} \cdot \boldsymbol{A}_{nn'} + Q_n^{t+1} V_n = 0 \tag{19}$$

In this work, we find the solution to the system of nonlinear equations given by
equation (19) using Newton's method with the Portable, Extensible Toolkit for
Scientific Computing (PETSc) library (Balay et al., 2016). PETSc provides a suite of
data structures and routines for the scalable solution of partial differential equations.
A Smooth approximation of the Brooks and Corey (1964) (SBC) water retention curve
was developed to facilitate faster convergence of the nonlinear solver (Appendix A).

### 2.3 VSFM single-column evaluation

We tested the VSFM with three idealized 1-dimensional test problems. First, the
widely studied problem for 1D Richards equation of infiltration in dry soil by Celia et
al. (1990) was used. The problem setup consists of a 1.0 m long soil column with a
uniform initial pressure of −10.0 m (= 3535.5 Pa). Time invariant boundary
conditions applied at the top and bottom of soil column are −0.75 m (= 9399.1 Pa)
and −10.0 m (= 3535.5 Pa), respectively. The soil properties for this test are given in
Table 1. A vertical discretization of 0.01 m is used in this simulation.

Second, we simulated transient one-dimensional vertical infiltration in two-
layered soil system as described in Srivastava and Yeh (1991). The domain consisted





of a 2 m tall soil column divided equally in two soil types. Except soil permeability, all
other soil properties of the two soil types are the same. The bottom soil is 10 times
less permeable than the top (Table1). Unlike Srivastava and Yeh (1991), who used
exponential functions of soil liquid pressure to compute hydraulic conductivity and
soil saturation, we used Mualem (1976) and van Genuchten (1980) constitutive
relationships. Since our choice of constitutive relationships for this setup resulted in
absence of an analytical solution, we compared VSFM simulations against PFLOTRAN
results. The domain was discretized in 200 control volumes of equal soil thickness.
Two scenarios, wetting and drying, were modeled to test the robustness of the VSFM
solver robustness. Initial conditions for each scenario included a time invariant
boundary condition of 0 m (= $1.01325 \times 10^5$ Pa) for the lowest control volume and a
constant flux of 0.9 cm hr$^{-1}$ and 0.1 cm hr$^{-1}$ at the soil surface for wetting and drying
scenarios, respectively.
Third, we compare VSFM and PFLOTRAN predictions for soil under variably
saturated conditions. The 1-dimensional 1 m deep soil column was discretized in 100
equal thickness control volumes. A hydrostatic initial condition was applied such that
water table is 0.5 m below the soil surface. A time invariant flux of $2.5 \times 10^{-5}$ m s$^{-1}$ is
applied at the surface, while the lowest control volume has a boundary condition
corresponding to the initial pressure value at the lowest soil layer. The soil properties
used in this test are the same as those used in the first evaluation.
**2.4   Global Simulations and groundwater depth analysis**
We performed global simulations with ELMv1-VSFM at a spatial resolution of
$1.9^0$ (latitude) × $2.5^0$ (longitude) with a 30 [min] time-step for 200 years, including a
180 year spinup and the last 20 years for analysis. The simulations were driven by
CRUNCEP meteorological forcing from 1991-2010 (Piao et al., 2012) and configured
to use prescribed satellite phenology.
For evaluation and calibration, we used the Fan et al. (2013) global ~1 km
horizontal resolution WTD dataset (hereafter F2013 dataset), which is based on a
combination of observations and hydrologic modeling. We aggregated the dataset to
the ELMv1-VSFM spatial resolution. ELM-VSFM's default vertical soil discretization





uses 15 soil layers to a depth of ~42 m, with an exponentially varying soil thickness.
However, ~13% of F2013 land gridcells have a water table deeper than 42 m. We
therefore modified ELMv1-VSFM to extend the soil column to a depth of 150 m with
59 soil layers; the first nine soil layer thicknesses were the same as described in
Oleson (2013) and the remaining layers (10-59) were set to a thickness of 3 m.

### 263    2.5   Estimation of the subsurface drainage parameterization

In the VSFM formulation, the dominant control on long-term GW depth is the
subsurface drainage flux, $q_d$ [kg m$^{-2}$ s$^{-1}$], which is calculated based on water table
depth, $z_\nabla$[m], (Niu et al. (2005)):

$$q_d = q_{d,max} exp(-f_d z_\nabla) \tag{20}$$

where $q_{d,max}$ [kg m$^{-2}$ s$^{-1}$] is the maximum drainage flux that depends on gridcell slope
and $f_d$ [m$^{-1}$] is an empirically-derived parameter. The subsurface drainage flux
formulation of Niu et al. (2005) is similar to the TOPMODEL formulation (Beven and
Kirkby, 1979) and assumes the water table is parallel to the soil surface. While
Sivapalan et al. (1987) derived $q_{d,max}$ as a function of lateral hydraulic anisotropy,
hydraulic conductivity, topographic index, and decay factor controlling vertical
saturated hydraulic conductivity, Niu et al. (2005) defined $q_{d,max}$ as a single
calibration parameter. ELMv0 uses $f_d = 2.5$ m$^{-1}$ as a global constant and estimates
maximum drainage flux when WTD is at the surface as $q_{d,max} = 10 \sin(\beta)$ kg m$^{-2}$ s$^-$
$^1$. Of the two parameters, $f_d$ and $q_{d,max}$, available for model calibration, we choose to
calibrate $f_d$ because the uncertainty analysis by Hou et al. (2012) identified it as the
most significant hydrologic parameter in CLM4. To improve on the $f_d$ parameter
values, we performed an ensemble of global simulations with $f_d$ values of 0.1, 0.2, 0.5,
1.0, 2.5, 5.0, 10.0, and 20 m$^{-1}$. Each ensemble simulation was run for 200 years to
ensure an equilibrium solution, and the last 20 years were used for analysis. A non-
linear functional relationship between $f_d$ and $WTD$ was developed for each gridcell
and then the F2013 dataset was used to estimate an optimal $f_d$ for each gridcell.



**2.6  Global ELM-VSFM evaluation**


With the optimal $f_d$ values, we ran a ELM-VSFM simulation using the protocol
described above. We then used the International Land Model Benchmarking package
(ILAMB) to evaluate the ELMv1-VSFM predictions of surface energy budget, total
water storage anomalies (TWSA), and river discharge. ILAMB evaluates model
prediction bias, RMSE, and seasonal and diurnal phasing against multiple
observations of energy, water, and carbon cycles at in-situ, regional, and global scales.
Since ELM-VSFM simulations in this study did not include an active carbon cycle, we
used the following ILAMB benchmarks for water and energy cycles: (i) latent and
surface energy fluxes using site-level measurements from FLUXNET (Lasslop et al.,
2010) and globally from FLUXNET-MTE (Jung et al., 2009)); (ii) terrestrial water
storage anomaly (TWSA) from the Gravity Recovery And Climate Experiment
(GRACE) observations (Kim et al., 2009); and (iii) stream flow for the 50 largest global
river basins (Dai and Trenberth, 2002). We applied ILAMB benchmarks for ELMv1-
VSFM simulations with default and calibrated $f_d$ to ensure improvements in WTD
predictions did not degrade model skill for other processes.

**3  Results and discussion**


**3.1  VSFM single-column evaluation**


For the 1D Richards equation infiltration in dry soil comparison, we evaluated
the solutions at 24-hr against those published by Celia et al. (1990) (Figure 1). The
VSFM solver accurately represented the sharp wetting front over time, where soil
hydraulic properties change dramatically due to non-linearity in the soil water
retention curve.
For the model evaluation of infiltration and drying in layered soil, the results of
the VSFM and PFLOTRAN are essentially identical. In both models and scenarios, the
higher permeability top soil responds rapidly to changes in the top boundary
condition and the wetting and drying fronts progressively travel through the less





permeable soil layer until soil liquid pressure in the entire column reaches a new
steady state by about 100 h (Figure 2).
We also evaluated the VSFM predicted water table dynamics against PFLOTRAN
predictions from an initial condition of saturated soil below 0.5 m depth. The
simulated water table rises to 0.3 m depth by 1 day and reaches the surface by 2 days,
and the VSFM and PFLOTRAN predictions are essentially identical Figure 3. These
three evaluation simulations demonstrate the VSFM accurately represents soil
moisture dynamics under conditions relevant to ESM-scale prediction.

### 319   3.2   Subsurface drainage parameterization estimation

The simulated nonlinear WTD-$f_d$ relationship is a result of the subsurface
drainage parameterization flux given by equation (20) (Figure 4(a) and (b)). For
$0.1 \leq f_d \leq 1$, the slope of the WTD-$f_d$ relationship for all gridcells is log-log linear
with a slope of $-1.0 \pm 0.1$. The log-log linear relationship breaks down for $f_d > 1$,
where the drainage flux becomes much smaller than infiltration and
evapotranspiration (Figure 4(c) and (d)). Thus, at larger $f_d$, the steady state $z_{\nabla}$
becomes independent of $f_d$ and is determined by the balance of infiltration and
evapotranspiration.
For 79% of the global gridcells, the ensemble range of simulated WTD spanned
the F2013 dataset. The optimal value of $f_d$ for each of these gridcells was obtained by
linear interpolation in the log-log space (e.g., Figure 4 (a)). For the remaining 21% of
gridcells where the shallowest simulated WTD across the range of $f_d$ was deeper than
that in the F2013 dataset, the optimal $f_d$ value was chosen as the one that resulted in
the lowest absolute WTD error (e.g., Figure 4 (b)). At large $f_d$ values, the drainage flux
has negligible effect on WTD, yet simulated WTD is not sufficiently shallow to match
the F2013 observations, which indicates that either evapotranspiration is too large
or infiltration is too small. There was no difference in the mean percentage of sand
and clay content between grids cells with and without an optimal $f_d$ value. The
optimal $f_d$ has a global average of 1.60 m$^{-1}$ $\pm$ 2.68 m$^{-1}$ and 72% of global gridcells have
an optimal $f_d$ value lower than the global average (Figure 5).



### 3.3 Global simulation evaluation

The ELMv1-VSFM predictions are much closer to the F2013 dataset (Figure 6a) using optimal globally-distributed $f_d$ values (Figure 6c) compared to the default $f_d$ value (Figure 6b). The significant reduction in WTD bias (model – observation) is mostly due to improvement in the model's ability to accurately predict deep WTD using optimal $f_d$ values. In the simulation using optimal globally-distributed $f_d$ values, all gridcells with WTD bias > 3.7 m were those for which an optimal $f_d$ was not found. The mean global bias, RMSE, and $R^2$ values were all improved in the new ELMv1-VSFM compared to the default model (Table 1). The 79% of global grid cells for which an optimal $f_d$ value was estimated had significantly better water table prediction with a bias, RMSE, and $R^2$ of -0.04 m, 0.67m, and 0.99, respectively, as compared to the remaining 21% of global grid that had a bias, RMSE, and $R^2$ of -9.82 m, 18.08m, and 0.31, respectively. The simulated annual WTD range, which we define to be the difference between maximum and minimum WTD in a year, has a spatial mean and standard deviation of 0.32 m and 0.58 m, respectively, using optimal $f_d$ values (Figure 7 (a)). The annual WTD range decreased by 0.24 m for the 79% of the grid cells for which an optimal $f_d$ value was estimated (Figure 7 (b)).

Globally-averaged WTD in ELMv1-VSFM simulations with default $f_d$ and optimal $f_d$ values were 10.5 m and 20.1 m, respectively. Accurate prediction of deep WTD in the simulation with optimal $f_d$ caused very small differences in near-surface soil moisture (Figure 8). The 79% of grid cells with an optimal $f_d$ value had deeper globally-averaged WTDs than when using the default $f_d$ value (24.3 m vs. 8.6 m). For these 79% of grid cells, the WTD was originally deep enough to not impact near-surface conditions (Kollet and Maxwell, 2008); therefore, further lowering of WTD led to negligible changes in near-surface hydrological conditions.

The International Land Model Benchmarking (ILAMB) package (Hoffman et al., 2017) provides a comprehensive evaluation of predictions of carbon cycle states and fluxes, hydrology, surface energy budgets, and functional relationships by comparison to a wide range of observations. We used ILAMB to evaluate the hydrologic and surface energy budget predictions from the new ELMv1-VSFM model



(Table 2). Optimal $f_d$ values had inconsequential impacts on simulated surface
energy fluxes at site-level and global scales. Optimal $f_d$ values led to improvement in
prediction of deep WTD (with a mean value of 24.3 m) for grid cells that had an
average WTD of 8.7 m in the simulation using default $f_d$ values. Thus, negligible
differences in surface energy fluxes between the two simulations are consistent with
the findings of Kollet and Maxwell (2008), who identified decoupling of groundwater
dynamics and surface processes at a WTD of ~10 m. There were slight changes in bias
and RMSE for predicted TWSA, but the ILAMB score remained unchanged. The TWSA
amplitude is lower for the simulation with optimal $f_d$ values, consistent with the
associated decrease in annual WTD range. ELM's skill in simulating runoff for the 50
largest global watersheds remained unchanged.

Finally, we evaluated the computational costs of implementing VSFM in ELM,

and compared them to the default model. We performed 5-years long simulations for
default and VSFM using 96, 192, 384, 768, and 1536 cores on the Edison
supercomputer at the National Energy Research Scientific Computing Center. Using
an optimal processor layout, we found that ELMv1-VSFM is ~30% more expensive
than the default ELMv1 model
**3.4   Caveats and Future Work**

The significant improvement in WTD prediction using optimal $f_d$ values

demonstrates VSFM's capabilities to model hydrologic processes using a unified
physics formulation for unsaturated-saturated zones. However, several caveats
remain due to uncertainties in model structure, model parameterizations, and climate
forcing data.

In this study, we assumed a spatially homogeneous depth to bedrock (DTB) of

150 m. Recently, Brunke et al. (2016) incorporated a global ~1 km dataset of soil
thickness and sedimentary deposits (Pelletier et al., 2016) in CLM4.5 to study the
impacts of soil thickness spatial heterogeneity on simulated hydrological and thermal
processes. While inclusion of heterogeneous DTB in CLM4.5 added more realism to
the simulation setup, no significant changes in simulated hydrologic and energy
fluxes were reported by Brunke et al. (2016). Presently, work is ongoing in the E3SM




project to include variable DTB within ELM and future simulations will examine the impact of those changes on VSFM's prediction of WTD. Our use of the 'satellite phenology' mode, which prescribes transient LAI profiles for each plant functional type in the gridcell, ignored the likely influence of water cycle dynamics and nutrient constraints on the C cycle (Ghimire et al., 2016; Zhu et al., 2016).

Lateral water redistribution impacts soil moisture dynamics (Bernhardt et al., 2012), biogeochemical processes in the root zone (Grant et al., 2015), distribution of vegetation structure (Hwang et al., 2012), and land-atmosphere interactions (Chen and Kumar, 2001; Rihani et al., 2010). The ELMv1-VSMF developed in this study does not include lateral water redistribution between soil columns and only simulates vertical water transport. Lateral subsurface processes can be included in LSMs via a range of numerical discretization approaches of varying complexity, e.g., adding lateral water as source/sink terms in the 1D model, implementing an operator split approach to solve vertical and lateral processes in a non-iterative approach (Ji et al., 2017), or solving a fully coupled 3D model (Bisht et al., 2017; Bisht et al., 2018; Kollet and Maxwell, 2008). Additionally, lateral transport of water can be implemented in LSMs at a subgrid level (Milly et al., 2014) or grid cell level (Miguez-Macho et al., 2007). The current implementation of VSFM is such that each processor solves the variably saturated Richards equation for all independent soil columns as one single problem. Thus, extension of VSFM to solve the tightly coupled 3D Richards equation on each processor locally while accounting for lateral transport of water within grid cells and among grid cells is straightforward. The current VSFM implementation can also be easily extended to account for subsurface transport of water among grid cells that are distributed across multiple processors by modeling lateral flow as a source/sink in the 1D model. Tradeoffs between approaches to represent lateral processes and computational costs need to be carefully studied before developing quasi or fully three-dimensional land surface models.

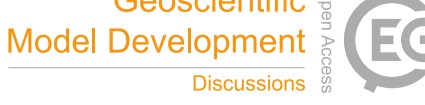



## 4    Summary and Conclusion


Starting from the climate-scale land model ELMv0, we incorporated a unified
physics formulation to represent soil moisture and groundwater dynamics that are
solved using PETSc. Application of VSFM to three benchmarks problems
demonstrated its robustness to simulated subsurface hydrologic processes in
coupled unsaturated and saturated zones. Ensemble global simulations at $1.9^0 \times 2.5^0$
were performed for 200 years to obtain spatially heterogeneous estimate of a
subsurface drainage parameter, $f_d$, that minimized the mismatch between predicted
and observed WTD. In order to simulated the deepest water table reported in the Fan
et al. (2013) dataset, we used 59 vertical soil layers that reached a depth of 150 m.
An optimal $f_d$ was obtained for 79% of the grids cells in the domain. For the
remaining 21% of grid cells, simulated WTD always remained deeper than observed.
Calibration of $f_d$ significantly improved global WTD prediction by reducing bias and
RMSE and increasing $R^2$. Grids without an optimal $f_d$ were the largest contributor of
error in WTD predication. ILAMB benchmarks on simulations with default and
optimal $f_d$ showed negligible changes to surface energy fluxes, TWSA, and runoff.
ILABM metrics ensured that model skill was not adversely impacted for all other
processes when optimal $f_d$ values were used to improve WTD prediction.

## 5    Appendix


### 5.1    Smooth approximation of Brooks-Corey water retention curve


The Brooks and Corey (1964) water retention curve of equation (10) has a
discontinuous derivative at $P = P_c^0$. Figure A 1 shown an example. To improve
convergence of the nonlinear solver at small capillary pressures, the smoothed
Brooks-Corey function introduces a cubic polynomial, $B(P_c)$, in the neighborhood of
$P_c^0$.

$$s_e = \begin{cases} (-\alpha P_c)^{-\lambda} & if\ P_c \leq P_u \\ B(P_c) & if\ P_u < P_c < P_s \\ 1 & if\ P_s \leq P_s \end{cases} \qquad (21)$$




where the breakpoints $P_u$ and $P_s$ satisfy $P_u < P_c^0 < P_s \leq 0$. The smoothing
polynomial

$$B(P_c) = b_0 + b_1(P_c - P_s) + b_2(P_c - P_s)^2 + b_3(P_c - P_s)^3 \tag{22}$$

introduces four more parameters, whose values follow from continuity. In particular
matching the saturated region requires $B(P_s) = b_0 = 1$, and a continuous derivative
at $P_c = P_s$ requires $B'(P_s) = b_1 = 0$. Similarly, matching the value and derivative at
$P_c = P_u$ requires

$$b_2 = \frac{-1}{\Delta^2}\left[3 - (\alpha P_u)^{-\lambda}\left(3 + \frac{\lambda\Delta}{P_u}\right)\right] \tag{23}$$

$$b_3 = \frac{-1}{\Delta^3}\left[2 - (\alpha P_u)^{-\lambda}\left(2 + \frac{\lambda\Delta}{P_u}\right)\right] \tag{24}$$

where $\Delta = P_u - P_s$. Note $P_u \leq \Delta < 0$.

In practice, setting $P_u$ too close to $P_c^0$ can produce an unwanted local maximum

in the cubic smoothing regime, resulting in se > 1. Avoiding this condition requires
that $B(P_c)$ increase monotonically from $P_c = P_u$, where $B'(P_c) > 0$, to $P_c = P_s$, where
$B'(P_c) = 0$. Thus a satisfactory pair of breakpoints ensures

$$B'(P_c) = [P_c - P_s][2b_2 + 3b_3(P_c - P_s)] > 0 \tag{25}$$

throughout $P_u \leq P_c < P_s$.

Let $P_c^*$ denote a local extremum of $B$, so that $B'(P_c^*) = 0$. If $P_c^* \neq P_s$, it follows

$P_c^* - P_s = -2b_2/(3b_3)$. Rewriting equation 22, $B'(P_c) = (P_c - P_s)3b_3(P_c - P_c^*)$ shows
that $B'(P_c^*) > 0$ requires either: (1) $b_3 < 0$ and $P_c^* < P_u$; or (2) $b_3 > 0$ and $P_c^* > P_u$;.
The first possibility places $P_c^*$ outside the cubic smoothing regime, and so does not
constrain the choice of $P_u$ or $P_s$. The second possibility allows an unwanted local
extremum at $P_u < P_c^* < P_s$. In this case, $b_3 > 0$ implies $b_2 < 0$ (since $P_c^* < P_s \leq 0$).
Then since $B''(P_c^*) = -2b_2$, the local extremum is a maximum, resulting in $s_e(P_c^*) >$

1.

Given a breakpoint $P_s$, one strategy for choosing $P_u$ is to guess a value, then

check whether the resulting $b_2$ and $b_3$ produces $P_u < P_c^* < P_s$. If so, $P_u$ should be
made more negative. An alternative strategy is to choose $P_u$ in order the guarantee
acceptable values for $b_2$ and $b_3$. One convenient choice forces $b_2 = 0$. Another picks
$P_u$ in order to force $b_3 = 0$. Both of these reductions: (1) ensure $B(P_c)$ has a positive



slope throughout the smoothing interval; (2) slightly reduce the computation cost of
finding $s_e(P_c)$ for $P_c$ on the smoothing interval; and (3) significantly reduce the
computational cost of inverting the model, in order to find $P_c$ as a function of $s_e$.
As shown in Figure A 1, the two reductions differ mainly in that setting $b_2 = 0$
seems to produce narrower smoothing regions (probably due to the fact that this
choice gives zero curvature at $P_c = P_s$, while $b_3 = 0$ yields a negative second
derivative there). However, we have not verified this observation analytically.
Both reductions require solving a nonlinear expression either equation (23) or
(24), for $P_u$. While details are beyond the scope of this paper, we note that we have
used a bracketed Newton-Raphson's method. The search switches to bisection when
Newton-Raphson would jump outside the bounds established by previous iterations,
and by the requirement $P_u < P_c^0$ In any event, since the result of this calculation may
be cached for use throughout the simulation, it need not be particularly efficient.

## 5.2   Residual equation of VSFM formulation

The residual equation for the VSFM formulation at $t + 1$ time level for $n$-th control
volume is given by

$$R_n^{t+1} \equiv \left( \frac{(\phi s_w \rho)_n^{t+1} - (\phi s_w \rho)_n^t}{\Delta t} \right) V_n + \sum_{n'} (\rho \boldsymbol{q})_{nn'}^{t+1} \cdot \boldsymbol{A}_{nn'} + Q_n^{t+1} V_n = 0 \qquad (26)$$

where $\phi$ [mm³ mm³] is the soil porosity, $s_w$ [-] is saturation, $\rho$ [kg m⁻³] is water
density, $\vec{q}_{nn'}$ [m s⁻¹] is the Darcy flow velocity between $n$-th and $n'$-th control
volumes, $A_{nn'}$ [m s⁻¹] is the interface face area between $n$-th and $n'$-th control
volumes $Q$ [kg m⁻³ s⁻¹] is a sink of water. The Darcy velocity is computed as

$$\boldsymbol{q}_{nn'} = -\left( \frac{k k_r}{\mu} \right)_{nn'} \left[ \frac{P_{n'} - P_n - \rho_{nn'}(\boldsymbol{g} \cdot \boldsymbol{d}_{nn'})}{d_n + d_{n'}} \right] \boldsymbol{n}_{nn'} \qquad (27)$$

where $\kappa$ [m⁻²] is intrinsic permeability, $\kappa_r$ [-] is relative permeability, $\mu$ [Pa s] is
viscosity of water, $P$ [Pa] is pressure], $\boldsymbol{g}$ [m s⁻²] is the acceleration due to gravity,
$d_n$ [m] and $d_{n'}$ [m] is distance between centroid of $n$-th and $n'$-th control volume to
the common interface between the two control volumes, $\boldsymbol{d}_{nn'}$ is a distance vector
joining centroid of $n$-th and $n'$-th control volume, and $\boldsymbol{n}_{nn'}$ is a unit normal vector
joining centroid of $n$-th and $n'$-th control volume.





The density at the interface of control volume, $\rho_{nn'}$, is computed as inverse
distance weighted average by

$$\rho_{nn'} = \omega_{n'}\rho_n + \omega_n\rho_{n'} \tag{28}$$

where $\omega_n$ and $\omega_{n'}$ are given by

$$\omega_n = \frac{d_n}{d_n + d_{n'}} = (1 - \omega_{n'}) \tag{29}$$

The first term on the RHS of equation 27 is computed as the product of distance
weighted harmonic average of intrinsic permeability, $k_{nn'}$, and upwinding of
$k_r/\mu\ (=\lambda)$ as

$$\left(\frac{kk_r}{\mu}\right)_{nn'} = k_{nn'}\left(\frac{k_r}{\mu}\right)_{nn'} = \left[\frac{k_n k_{n'}(d_n + d_{n'})}{k_n d_{n'} + k_{n'} d_n}\right]\lambda_{nn'} \tag{30}$$

where

$$\lambda_{nn'} = \begin{cases} (k_r/\mu)_n & if\ \vec{q}_{nn'} > 0 \\ (k_r/\mu)_{n'} & otherwise \end{cases} \tag{31}$$

By substituting equation 28, 29 and 30 in equation 27, we obtain

$$\boldsymbol{q}_{nn'} = -\left[\frac{k_n k_{n'}}{k_n d_{n'} + k_{n'} d_n}\right]\lambda_{nn'}[P_{n'} - P_n - \rho_{nn'}(\boldsymbol{g}.\boldsymbol{d}_{nn'})]\boldsymbol{n}_{nn'} \tag{32}$$


### 5.3   Jacobian equation of VSFM formulation

The discretized equations of VSFM leads to a system of nonlinear equations given by
$\boldsymbol{R}^{t+1}(\boldsymbol{P}^{t+1}) = \boldsymbol{0}$, which are solved using Newton's method using the Portable,
Extensible Toolkit for Scientific Computing (PETSc) library. The algorithm of
Newton's method requires solution of the following linear problem

$$\boldsymbol{J}^{t+1,k}(\boldsymbol{P}^{t+1,k})\,\Delta\boldsymbol{P}^{t+1,k} = -\boldsymbol{R}^{t+1,k}(\boldsymbol{P}^{t+1,k}) \tag{33}$$

where $\boldsymbol{J}^{t+1,k}(\boldsymbol{P}^{t+1,k})$ is the Jacobian matrix. In VSFM, the Jacobian matrix is
computed analytically. The contribution to the diagonal and off-diagonal entry of the
Jacobian matrix from $n$-th residual equations are given by

$$J_{nn} = \frac{\partial R_n}{\partial P_n} = \left(\frac{V_n}{\Delta t}\right)\frac{\partial(\rho\phi s_w)}{\partial P_n} + \sum_{n'}\frac{\partial(\rho\boldsymbol{q})_{nn'}}{\partial P_n}A_{nn'} + \frac{\partial Q_n^{t+1}}{\partial P_n}V_n \tag{34}$$





$$J_{nn'} = \frac{\partial R_n}{\partial P_{n'}} = \sum_{n'} \frac{\partial (\rho \boldsymbol{q})_{nn'}}{\partial P_{n'}} \boldsymbol{A}_{nn'} + \frac{\partial Q_n^{t+1}}{\partial P_{n'}} V_n \tag{35}$$

The derivative of the accumulation term in $J_{nn}$ is computed as

$$\frac{\partial (\rho \phi s_w)}{\partial P_n} = \phi s_w \frac{\partial \rho}{\partial P_n} + \rho s_w \frac{\partial \phi}{\partial P_n} + \rho \phi \frac{\partial s_w}{\partial P_n} \tag{36}$$

The derivative of flux between $n$-th and $n'$-th control volume with respect to
pressure of each control volume is given as

$$\frac{\partial (\rho \boldsymbol{q})_{nn'}}{\partial P_n} = \rho_{nn'} \frac{\partial \boldsymbol{q}_{nn'}}{\partial P_n} + \boldsymbol{q}_{nn'} \omega_n \frac{\partial \rho_n}{\partial P_n} \tag{37}$$


$$\frac{\partial (\rho \boldsymbol{q})_{nn'}}{\partial P_{n'}} = \rho_{nn'} \frac{\partial \boldsymbol{q}_{nn'}}{\partial P_{n'}} + \boldsymbol{q}_{nn'} \omega_{n'} \frac{\partial \rho_{n'}}{\partial P_{n'}} \tag{38}$$

Lastly, the derivative of Darcy velocity between $n$-th and $n'$-th control volume with
respect to pressure of each control volume is given as

$$\frac{\partial \boldsymbol{q}_{nn'}}{\partial P_n} = \left[ \frac{k_n k_{n'}}{k_n d_{n'} + k_{n'} d_n)} \right] \lambda_{nn'} \left[ 1 + \omega_n (\boldsymbol{g} \cdot \boldsymbol{d}_{nn'}) \frac{\partial \rho_n}{\partial P_n} \right] \boldsymbol{n}_{nn'} + \boldsymbol{q}_{nn'} \frac{\partial \big( ln(\lambda_{nn'}) \big)}{\partial P_n} \tag{39}$$

$$\frac{\partial \boldsymbol{q}_{nn'}}{\partial P_{n'}} = \left[ \frac{k_n k_{n'}}{k_n d_{n'} + k_{n'} d_n)} \right] \lambda_{nn'} \left[ -1 + \omega_n (\boldsymbol{g} \cdot \boldsymbol{d}_{nn'}) \frac{\partial \rho_{n'}}{\partial P_{n'}} \right] \boldsymbol{n}_{nn'}$$
$$+ \boldsymbol{q}_{nn'} \frac{\partial \big( ln(\lambda_{nn'}) \big)}{\partial P_{n'}} \tag{40}$$


## 6    Code availability

The standalone VSFM code is available at https://github.com/MPP-LSM/MPP. The
ELM-VSFM code will be made available with the public release of E3SM model in
April, 2018.

## 7    Competing interests

The authors declare that they have no conflict of interest.




## 8   Acknowledgements

This research was supported by the Director, Office of Science, Office of Biological
and Environmental Research of the US Department of Energy under contract no. DE-
AC02-05CH11231 as part of the Energy Exascale Earth System Model (E3SM)
programs.




## 541   9   Tables

**542   Table 1 Bias, root mean square error (RMSE), and correlation (R²) between**

**543   simulated water table depth and Fan et al. (2013) data.**

|  | Bias [m] | RMSE [m] | $R^2$ |
|---|---|---|---|
| For all grids in ELM simulation with default $f_{drain}$ | -10.3 | 21.3 | 0.28 |
| For all grids in ELM simulation with optimal $f_{drain}$ | 2.10 | 8.33 | 0.91 |
| For 79% grids with optimal $f_{drain}$ in ELM simulation with optimal $f_{drain}$ | -0.04 | 0.67 | 0.99 |
| For 21% grids without optimal $f_{drain}$ in ELM simulation with optimal $f_{drain}$ | -9.82 | 18.08 | 0.31 |





**Table 2 ILAMB benchmark scores for latent heat flux (LH), sensible heat flux**
**(SH), total water storage anomaly (TWSA), and surface runoff. The calculation**
**of ILAMB metrics and scores are described at http://redwood.ess.uci.edu/.**

| | Data Source | Simulation with default $f_d$ | | | Simulation with optimal $f_d$ | | |
|---|---|---|---|---|---|---|---|
| | | **Bias** | **RMSE** | **ILAMB Score** | **Bias** | **RMSE** | **ILAMB Score** |
| LH | FLUXNET | 10.1 [Wm$^{-2}$] | 21.0 [Wm$^{-2}$] | 0.68 | 9.5 [Wm$^{-2}$] | 21.3 [Wm$^{-2}$] | 0.68 |
| | GBAF | 7.1 [Wm$^{-2}$] | 16.3 [Wm$^{-2}$] | 0.81 | 6.3 [Wm$^{-2}$] | 16.3 [Wm$^{-2}$] | 0.81 |
| SH | FLUXNET | 6.7 [Wm$^{-2}$] | 22.5 [Wm$^{-2}$] | 0.66 | 7.1 [Wm$^{-2}$] | 22.8 [Wm$^{-2}$] | 0.65 |
| | GBAF | 6.9 [Wm$^{-2}$] | 21.2 [Wm$^{-2}$] | 0.71 | 7.6 [Wm$^{-2}$] | 21.7 [Wm$^{-2}$] | 0.70 |
| TWSA | GRACE | 1.3 [cm] | 7.8 [cm] | 0.48 | 3.0 [cm] | 9.6 [cm] | 0.48 |
| Runoff | Dai | -0.26 [kg $^{m-2}$ d$^{-1}$] | 0.91 [m$^{-2}$m$^{-2}$ d$^{-1}$] | 0.52 | -0.23 [kg m$^{-2}$ d$^{-1}$] | 0.88 [kg m$^{-2}$ d$^{-1}$] | 0.50 |



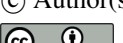



**10 Figures**

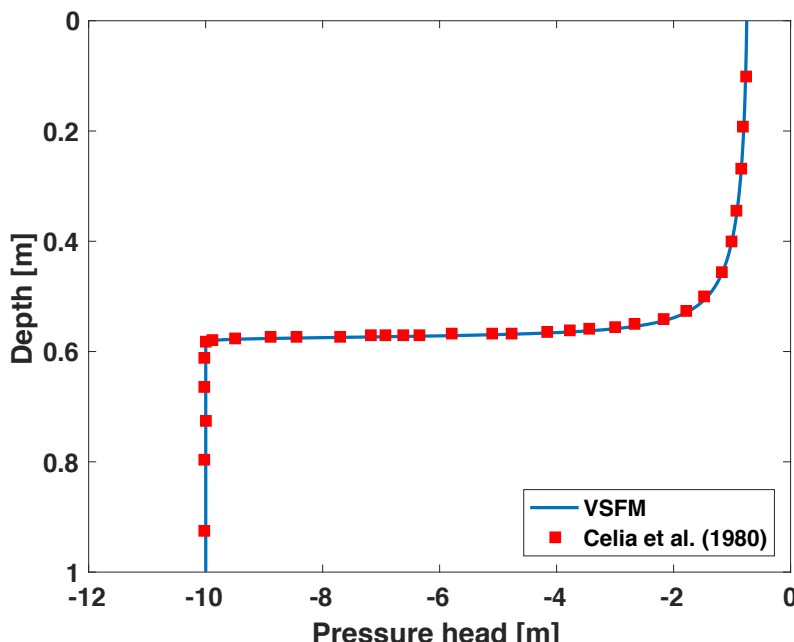


**Figure 1. Comparison of VSFM simulated pressure profile (blue line) against**
**data (red square) reported in Celia et al. (1990) at time = 24 hr for infiltration**
**in a dry soil column. Initial pressure condition is shown by green line.**



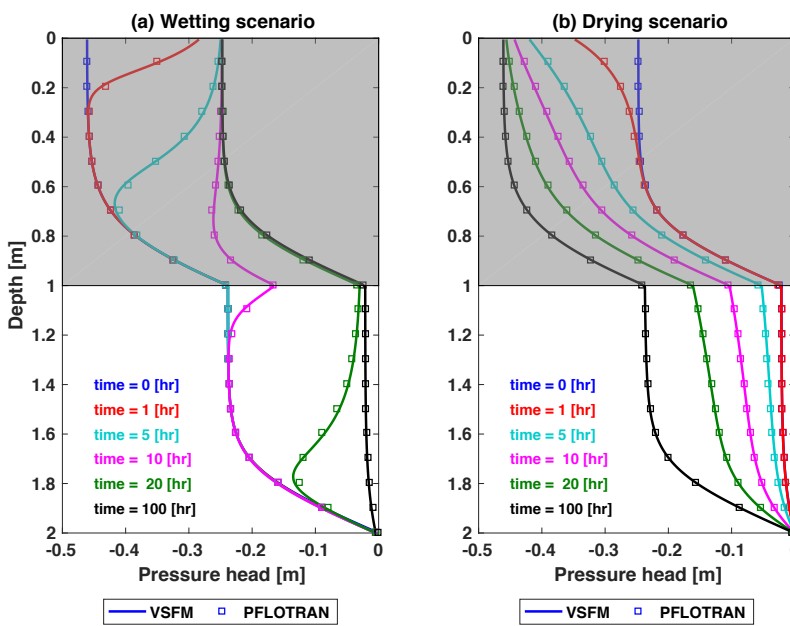


**Figure 2. Transient liquid pressure simulated for a two layer soil system by
VSFM (solid line) and PFLOTRAN (square) for wetting (left) and drying (right)
scenarios.**



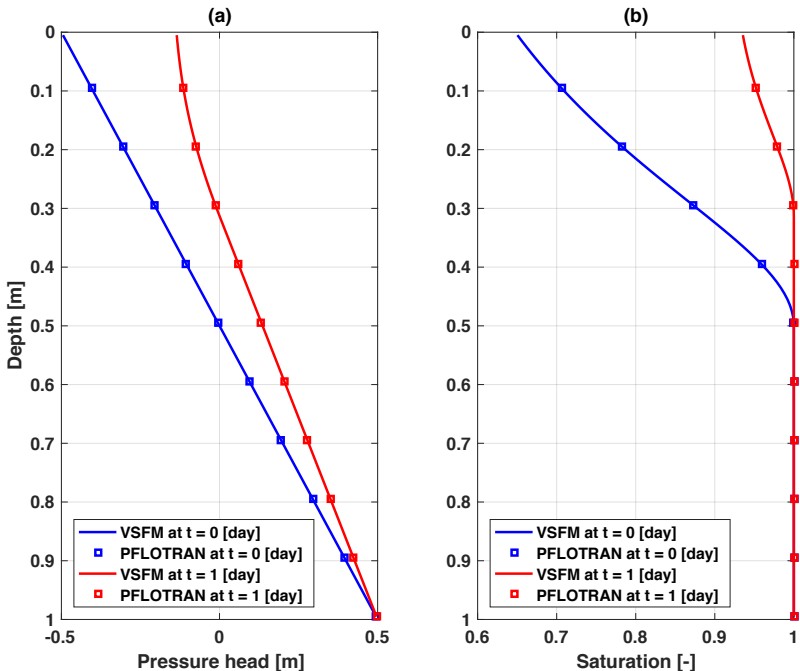


**Figure 3. Transient liquid pressure (a) and soil saturation (b) simulated by
VSFM (solid line) and PFLOTRAN (square) for the water table dynamics test
problem.**



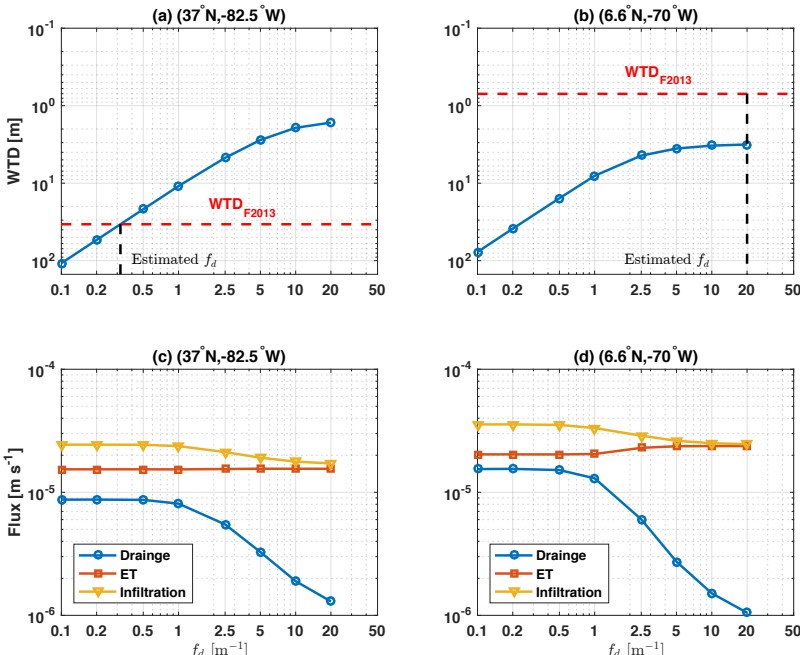


**Figure 4. (a-b) The nonlinear relationship between simulated water table depth (WTD) and $f_d$ for two gridcells within ELM's global grid. WTD from the Fan et al. (2013) dataset and optimal $f_d$ for the two gridcells are shown with a dashed red and dashed black lines, respectively. (c-d) The simulated drainage, evapotranspiration, and infiltration fluxes as functions of optimal $f_d$ for the two ELM gridcells.**






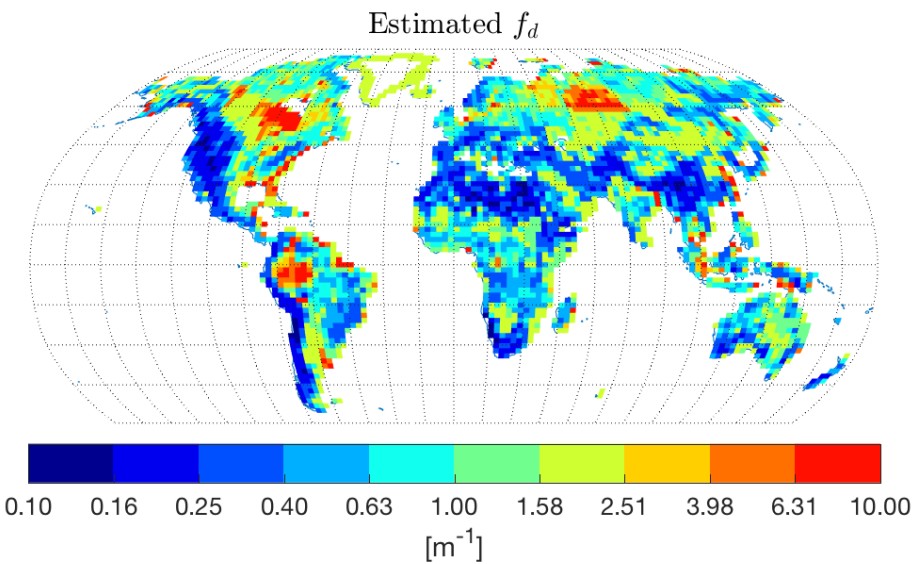


**Figure 5. Global estimate of $f_d$.**

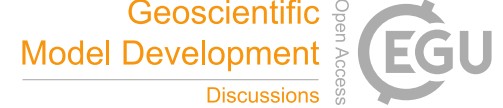



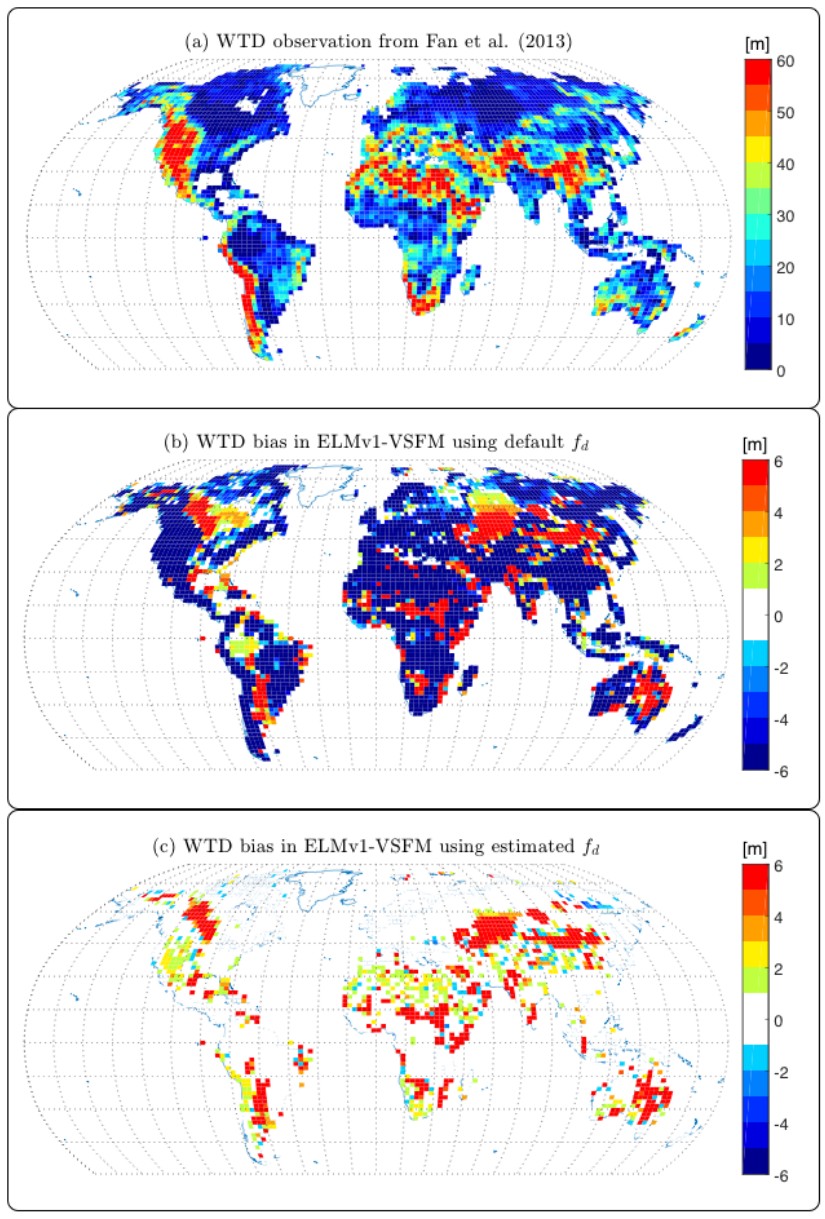


**Figure 6. (a) Water table depth observation from Fan et al. (2013); (b) Water table depth biases (=Model – Obs) from ELMv1-VSFM using default spatially homogeneous $f_d$; and (c) Water table depth biases from ELMv1-VSFM using spatially heterogeneous $f_d$.**




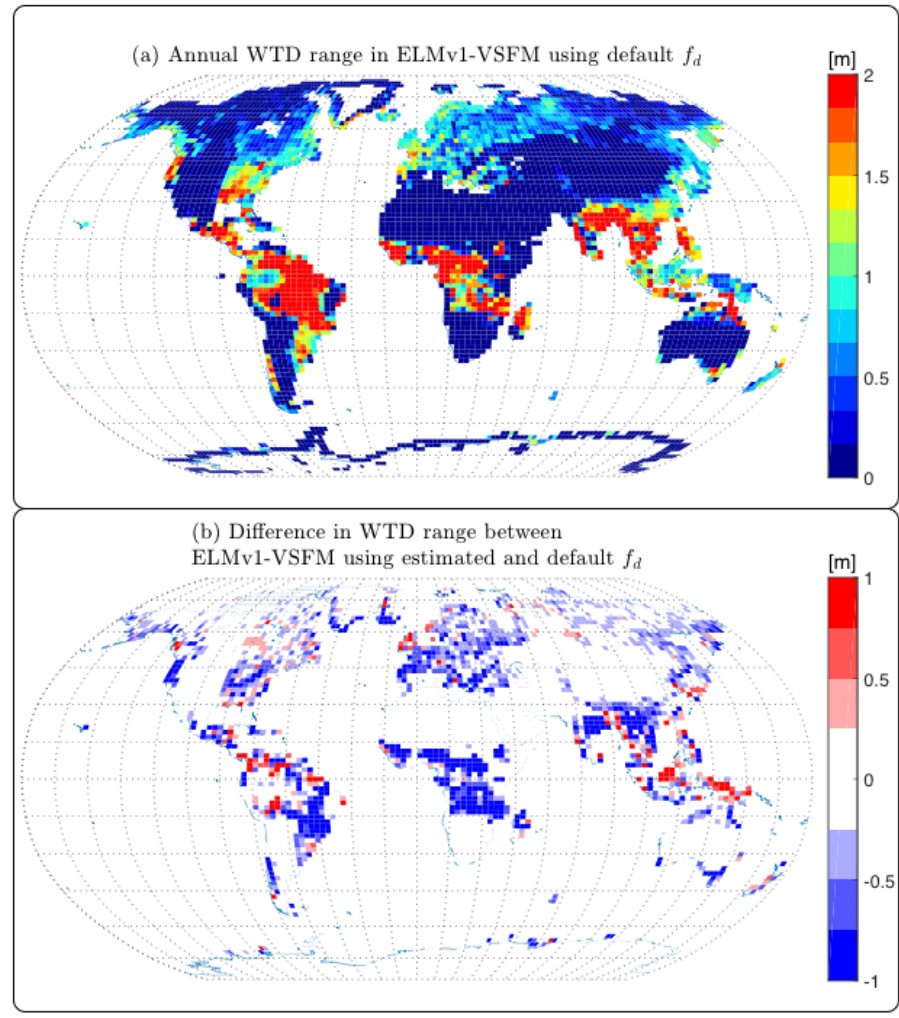


**Figure 7. (a) Annual range of water table depth for ELMv1-VSFM simulation**
**with spatially heterogeneous estimates of $f_d$ and (b) Difference in annual**
**water table depth range between simulations with optimal and default $f_d$.**





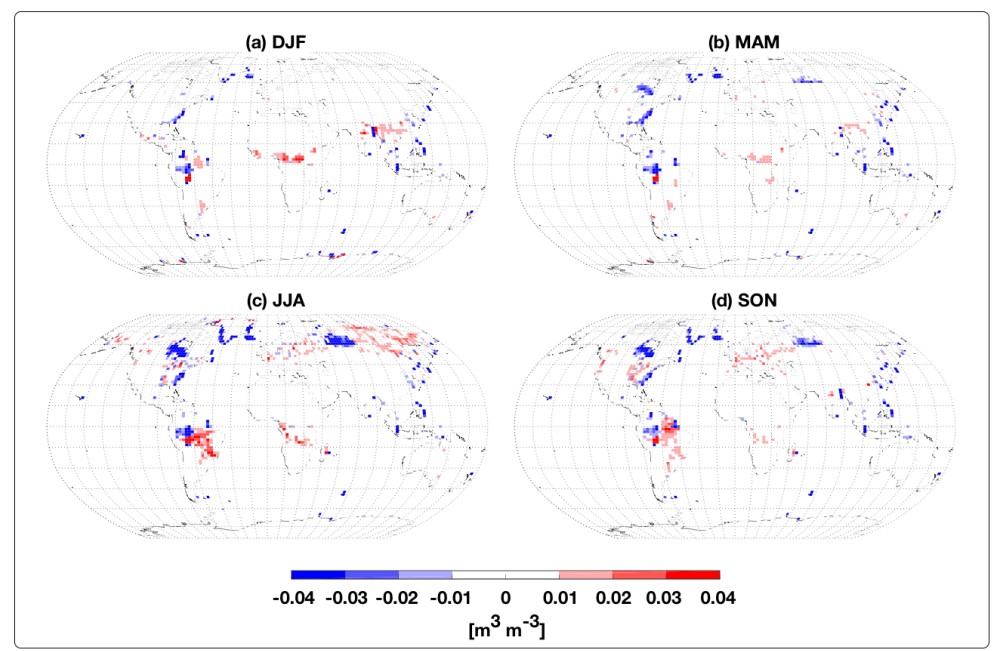


**Figure 8. Seasonal monthly mean soil moisture differences for top 10 cm between ELMv1-VSFM simulations with optimal and default $f_d$ values.**







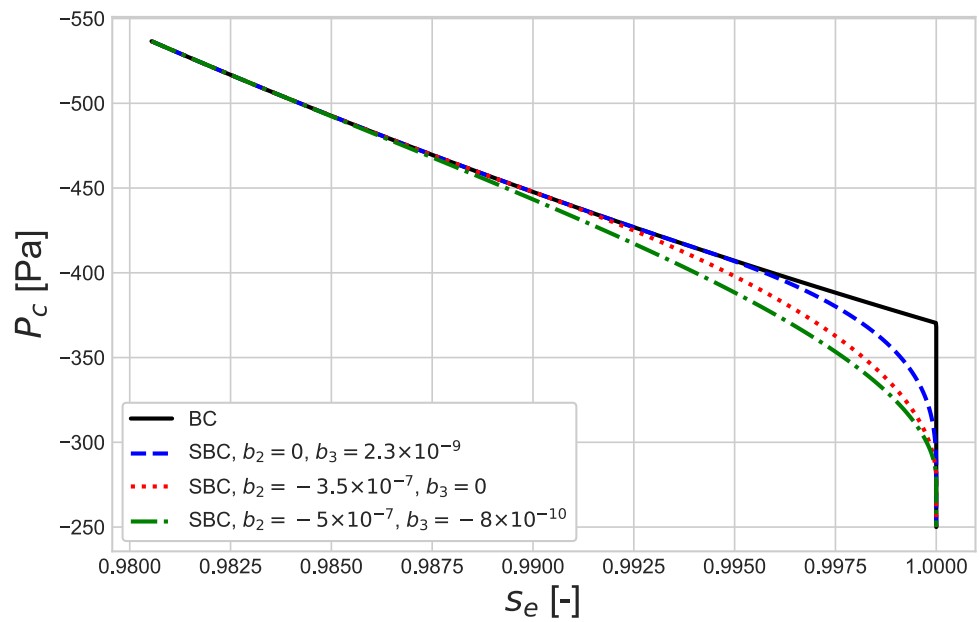


**Figure A 1 The Brooks-Corey water rendition curve for estimating liquid saturation, $s_e$,**
**as a function of capillary pressure, $P_c$, shown in solid black line and smooth**
**approximation of Brooks-Corey (SBC) are shown in dashed line.**





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
