# Peer review of "Development and evaluation of a variably saturated flow model in the global E3SM Land Model (ELM) Version 1.0 Gautam Bisht1, William J. Riley1, Glenn E. Hammond2, and David M. Lorenzetti3 1Climate & Ecosystem Sciences Division, Lawrence Berkeley"

_Geoscientific Model Development, 2018_

## Short Comment (SC1) · 15 Mar 2018

To maintain reproducibility the authors need to tag the release in the GitHub repository that matches the model version described in the paper. This grantees that check-ins after manuscript submission don't compromise reproducibility. As explained in https://www.geoscientific-model-development.net/about/manuscript_types.html the preferred reference to this release is through the use of a DOI which is then cited in the paper. For projects in GitHub a DOI for a released code version can easily be created using for instance Zenodo, see https://guides.github.com/activities/citable-code/ for details.

The authors need also clarify how and under what conditions the reader can access

the relevant version of the E3SM code. It is expected that any program code used for the results presented in the paper is available at the point of manuscript submission. Reference to a later release is not within the guidelines of GMD.

Lutz Gross GMD Executive Editor

---

## Referee Comment (RC1) · Anonymous Referee #1 · 30 Mar 2018

Bisht et al. developed and evaluated a one dimensional variably saturated flow model (VSFM) and calibrated spatially heterogeneous subsurface drainage parameters for ELM. They were able to significantly improve water table depth prediction using this model. I believe the major contribution of this work is the calibrated drainage parameters. Overall, the manuscript is not quite well organized or written. I couldn't find a motivation in the manuscript why one wants to spend extra 30% computation time using VSFM. The existing flow formulation was described, but the model was only compared against PFLOTRAN. How does it compare to the existing formulation in ELM? Would ELM perform equally well using the existing flow formulation with the new drainage parameters?

[Figure]

Specific comments:

1. There is a great deal of efforts describing different models and the importance of groundwater system in the introduction, but no justification of why a new model is in need.

2. Eq. (6), missing "z" in the term after the second "=".

3. Eq. (10, "P" in the second if should be "Pc".

4. Eq. (14), missing dV in the last term. I didn't go through all the equations, but the authors should check for correctness/completeness of each one of them, including the appendix.

5. Make sure every variable in the equations is defined. For example, what's T in Eq.(13)?

6. Page 10, line 223: correct the conversion as -0.75 m is not equivalent to 9399.1 Pa.

7. Table 1 mentioned on page 10, line 225 is missing.

8. Figure 1 – where is the green line?

9. Figure 4 – which one is a,b,c,or d?
* * *

---

## Referee Comment (RC2) · Anonymous Referee #2 · 2 Jun 2018

Strength 1. The simulations derived from ELMv1-VSFM is compared with the different dataset and other simulations so that the performance of the model is expressed well. 2. They developed a method for subsurface drainage parameterization and its performance in estimating WTD in global scale is analyzed.

Weakness 1. Given that ELMv1-VSFM is ∼30% more expensive than the default ELMv1 model, the advantage of using a unified physics formulation is not clearly indicated in the paper. 1.1 The reason why they intended to unify the treatment of soil hydrologic processes should be stated. (c.f. Distinct representation for different flow domains, unsaturated zone, and aquifer, is more useful to represent dynamic interac-

tions between the flow domains) 2. They use variably saturated Richards' equation to estimate WTD using the relationship of soil moisture-pressure head. Assuming proper soil type for each soil layer is critical for accurately estimating pressure distribution in the soil column, but their assumption (or investigation) about soil type in the soil column is not indicated. 3. Lateral movements in phreatic zone (aquifer) is not considered. The variably saturated Richards' equation is the form that can apply to 3-dimensional analysis, but this study uses the equation only for vertical flow in the soil column. 4. Why is a zero-flux boundary condition applied to the last hydrologically active soil layer when the water table is within the soil column? 4.1 Constant pressure head condition could be used as a bottom boundary condition to represent the water table located within the soil column. 5. The simulations derived from the newly developed model VSFM are evaluated by comparing with the simulations from PFLOTRAN. As a supplementary basis for model prediction performance, the authors use ILAMB score. For the details of ILAMB metrics and scores are not indicated on the paper, it is difficult to determine the performance of the model prediction skill with ILAMB score (How the ILAMB provides a comprehensive evaluation of predictions of carbon cycle states and fluxes, hydrology, surface energy should be stated). 6. The area of each grid-cell of ELMv1-VSFM is 1.90 (latitude) $\times$ 2.50 (longitude) and time-step is 30 min. Some indexes could be used to show that the variably saturated Richards' equation is converged well in that spatio-temporal scale (e.g., Peclet number). Plus, if the authors indicate what method (e.g., upwind difference scheme) they use to determine interfacial properties (e.g., hydraulic conductivity), their work will be better understood by readers.

---

## Author Comment (AC1) · 26 Jun 2018

**Development and evaluation of a variably saturated flow model in the global E3SM Land Model (ELM) Version 1.0 [MS no. gmd-2018-44]**

**SC1: 'Code Availability', Lutz Gross**

To maintain reproducibility the authors need to tag the release in the GitHub repository that matches the model version described in the paper. This grantees that check-ins after manuscript submission don't compromise reproducibility. As explained in https://www.geoscientific-model-development.net/about/manuscript_types.html the preferred reference to this release is through the use of a DOI which is then cited in the paper. For projects in GitHub a DOI for a released code version can easily be created using for instance Zenodo, see https://guides.github.com/activities/citable-code/ for details. The authors need also clarify how and under what conditions the reader can access the relevant version of the E3SM code. It is expected that any program code used for the results presented in the paper is available at the point of manuscript submission. Reference to a later release is not within the guidelines of GMD

**Response:**

The research was performed using E3SM v1.0 and the code is available at

https://github.com/E3SM-Project/E3SM

E3SM Project, DOE. Energy Exascale Earth System Model. Computer Software. https://github.com/E3SM-Project/E3SM.git. 23 Apr. 2018. Web. doi:10.11578/E3SM/dc.20180418.36. We have added this information to Introduction section (line 131-142).

---

## Author Comment (AC2) · 26 Jun 2018

**Development and evaluation of a variably saturated flow model in the global E3SM Land Model (ELM) Version 1.0 [MS no. gmd-2018-44]**

**RC1: 'Review of the manuscript by Bisht et al.', Anonymous Referee #1**

Bisht et al. developed and evaluated a one dimensional variably saturated flow model (VSFM) and calibrated spatially heterogeneous subsurface drainage parameters for ELM. They were able to significantly improve water table depth prediction using this model. I believe the major contribution of this work is the calibrated drainage parameters.

Overall, the manuscript is not quite well organized or written. I couldn't find a motivation in the manuscript why one wants to spend extra 30% computation time using VSFM.
Why is the new model justified given 30% computational cost?

**Response:**

We have updated sections 2.2.1, 3.4, and 3.5 to highlight the features of the VSFM model and justified the increase in 30% computational cost. Here is a summary of those modifications:

- The modular software design of VSFM allows it to be built independently of the ELM code. This flexibility of the VSFM build system allows testing of the model's physics without any influence from the rest of ELM's physics formulations. Additionally, the modular software design of VSFM does not limit its application to a problem with only a fixed boundary and source-sink conditions. VSFM can be easily configured for a problem with different types of spatial grid resolutions, material properties, boundary conditions, and source-sink terms. The previous version of ELM did not allow for this flexibility.

- VSFM uses PETSc's DMComposite capability that adds flexibility for solving tightly coupled multi-component problems (e.g., transport of water through the soil-plant continuum) and multi-physics problems (e.g., fully coupled conservation of mass and energy equations in the subsurface). The previous version of ELM was unable to solve these types of problems without extensive modification.

- The relative computational cost of the land model in a fully coupled global model simulation is very low. Dennis et al. (2012) reported computational cost of the land

model to be less than 1% in ultra-high-resolution CESM simulations. Thus, the increase of 30% computational cost of ELM is expected to be not very significant within fully coupled E3SM simulation, and we argue the enhancements described in the current paper far outweigh the modest increased computational cost.

The existing flow formulation was described, but the model was only compared against PFLOTRAN. How does it compare to the existing formulation in ELM?

**Response:**

ELMv0 code for subsurface hydrologic processes only supports two vertical mesh configurations and a single set of boundary and source-sink conditions. The mesh configurations and boundary conditions required for solving benchmark problems is unsupported by ELMv0. Moreover, the monolithic ELMv0 code does not allow for testing of individual process representations against analytical solutions or simulation results from other models. We have updated text in Section 2.3 to include these reasons why comparison of VSFM was not performed against ELMv0 for the multiple benchmark problems. One of the abilities of the new model is its easy configurability for benchmarking across a wide range of problems. We have added extensive notes on how to run the VSFM for all benchmark problems and compare results against PFLOTRAN at [https://bitbucket.org/gbisht/notes-for-gmd-2018-44](https://bitbucket.org/gbisht/notes-for-gmd-2018-44). Additionally, we have updated the code availability section to include the above-mentioned notes for reproducing our results for the benchmark problems.

Would ELM perform equally well using the existing flow formulation with the new drainage parameters?

**Response:**

ELM's existing saturated zone flow formulation, which is based on the unconfined aquifer model of Niu et al. (2007), is only setup to simulate a maximum WTD of 42.1 [m]. Thus, ELM's existing saturated zone flow is incapable of accurately simulating WTD for the ~13% of global grid cells that have a water table deeper than 42 [m] (Fan et al. (2013). While extending the Niu et al. (2007) unconfined aquifer model for grid cells with WTD greater than 42 [m] and estimating optimized drainage parameters is beyond the scope of this work.

Specific comments:

1. There is a great deal of efforts describing different models and the importance of groundwater system in the introduction, but no justification of why a new model is in need.

**Response:**

The introduction has been updated to include reference to the Clark et al. (2015) study that summarized the lack of unified treatment of soil hydrologic processes in current generation LSMs and identified incorporation of a variably saturated Richards' model in future LSMs as a key modeling development opportunity.

2. Eq. (6), missing "z" in the term after the second "=".

**Response:**

Equation 6 has been updated to include the missing term.

3. Eq. (10, "P" in the second if should be "Pc".

**Response:**

Equation 10 has been updated to use $P_c$ instead of $P$.

4. Eq. (14), missing dV in the last term. I didn't go through all the equations, but the authors should check for correctness/completeness of each one of them, including the appendix.

**Response:**

The missing $dV$ in the third term of equation 14 has been added.

5. Make sure every variable in the equations is defined. For example, what's T in Eq.(13)?

**Response:**

We have updated the description on line191-192 to include the definition of $T$ (i.e., soil temperature). We have also gone through the entire manuscript to ensure that all variables are defined in the text.

6. Page 10, line 223: correct the conversion as -0.75 m is not equivalent to 9399.1 Pa.

**Response:**

We corrected the equivalent head of -0.75 m to 93989.1 Pa.

7. Table 1 mentioned on page 10, line 225 is missing.

**Response:**

The missing table containing parameters for the benchmark problems is included now.

8. Figure 1 – where is the green line?

**Response:**

The figure has been updated to include the initial pressure profile by a green line.

9. Figure 4 – which one is a,b,c,or d?

**Response:**

The title of subplots in Figure 4 do include a, b, c and d.

**References**

Clark, M. P., Fan, Y., Lawrence, D. M., Adam, J. C., Bolster, D., Gochis, D. J., Hooper, R. P., Kumar, M., Leung, L. R., Mackay, D. S., Maxwell, R. M., Shen, C., Swenson, S. C., and Zeng, X.: Improving the representation of hydrologic processes in Earth System Models, Water Resources Research, 51, 5929-5956, 2015.

Dennis, J. M., Vertenstein, M., Worley, P. H., Mirin, A. A., Craig, A. P., Jacob, R., and Mickelson, S.: Computational performance of ultra-high-resolution capability in the Community Earth System Model, The International Journal of High Performance Computing Applications, 26, 5-16, 2012.

Fan, Y., Li, H., and Miguez-Macho, G.: Global Patterns of Groundwater Table Depth, Science, 339, 940-943, 2013.

Niu, G.-Y., Yang, Z.-L., Dickinson, R. E., Gulden, L. E., and Su, H.: Development of a simple groundwater model for use in climate models and evaluation with Gravity Recovery and Climate Experiment data, Journal of Geophysical Research: Atmospheres, 112, n/a-n/a, 2007.

---

## Author Comment (AC3) · 26 Jun 2018

**Development and evaluation of a variably saturated flow model in the global E3SM Land Model (ELM) Version 1.0 [MS no. gmd-2018-44]**

**RC2: 'Confidential Review A review for Development and evaluation of a variably saturated flow model in the global E3SM', Anonymous Referee #2**

Strength 1. The simulations derived from ELMv1-VSFM is compared with the different dataset and other simulations so that the performance of the model is expressed well. 2. They developed a method for subsurface drainage parameterization and its performance in estimating WTD in global scale is analyzed.

Weakness

1. Given that ELMv1-VSFM is ~30% more expensive than the default ELMv1 model, the advantage of using a unified physics formulation is not clearly indicated in the paper.

**Response:**

As discussed in our responses to Reviewer #1, we have updated sections 2.2.1, 3.4, and 3.5 to highlight the features of VSFM model and justified the increase in 30% computational cost. Please see above for a summary of those modifications.

1.1 The reason why they intended to unify the treatment of soil hydrologic processes should be stated. (c.f. Distinct representation for different flow domains, unsaturated zone, and aquifer, is more useful to represent dynamic interactions between the flow domains)

**Response:**

The introduction has been updated to include reference to the Clark et al. (2015) study that summarized the lack of unified treatment of soil hydrologic processes in current generation LSMs and identified incorporation of a variably saturated Richards' model in future LSMs as a key modeling development opportunity.

2. They use variably saturated Richards' equation to estimate WTD using the relationship of soil moisture-pressure head. Assuming proper soil type for each soil layer is critical for

accurately estimating pressure distribution in the soil column, but their assumption (or investigation) about soil type in the soil column is not indicated.

**Response:**

We used the same characterization of soil properties that is used in the baseline ELMv0 model. We have added a sentence in section 3.4 describing this soil characterization in VSFM.

3. Lateral movements in phreatic zone (aquifer) is not considered. The variably saturated Richards' equation is the form that can apply to 3-dimensional analysis, but this study uses the equation only for vertical flow in the soil column.

**Response:**

In section 3.4, we did discuss possible approaches to extend the current 1-dimensional, vertical-only ELMv1-VSFM to include lateral flows with a range of model complexity (i.e. 1D model with source/sink term, 1D model coupled non-iteratively to 2D model or full 3D). Clark et al. (2015) acknowledged that while the need to incorporate lateral flow is well understood, the most effective modeling approach for global LSMs to include lateral subsurface flow is unclear. The development of unified treatment of hydrologic processes in unsaturated and saturated zone in this research is the first step towards incorporating lateral flows in future versions of ELM.

4. Why is a zero-flux boundary condition applied to the last hydrologically active soil layer when the water table is within the soil column? 4.1 Constant pressure head condition could be used as a bottom boundary condition to represent the water table located within the soil column.

**Response:**

ELMv0 applies a zero-flux boundary condition to the last hydrologically active soil layer when the water table is within the soil column, while VSFM uses a zero-flux boundary condition for the last soil layer. Our first version of the manuscript omitted a description of this lower boundary condition, which we have rectified on line 208 of the revised manuscript.

5. The simulations derived from the newly developed model VSFM are evaluated by comparing with the simulations from PFLOTRAN. As a supplementary basis for model prediction performance, the authors use ILAMB score. For the details of ILAMB metrics and scores are not indicated on the paper, it is difficult to determine the performance of the model prediction skill with ILAMB score (How the ILAMB provides a comprehensive evaluation of predictions of carbon cycle states and fluxes, hydrology, surface energy should be stated).

**Response:**

To address the reviewer's comment regarding details on ILAMB metrics we have cited two publications describing the ILAMB benchmarking package (Collier et al., 2018; Hoffman et al., 2017). Our intent with discussing the ILAMB benchmarking results in the current manuscript is to indicate that very small differences existed from the baseline metrics used globally to evaluate the hydrological components of the simulations (i.e., surface energy balance, runoff, total water storage anomaly). To clarify this goal, we have revised the text on lines 311-312.

6. The area of each grid-cell of ELMv1-VSFM is 1.90 (latitude) × 2.50 (longitude) and time-step is 30 min. Some indexes could be used to show that the variably saturated Richards' equation is converged well in that spatiotemporal scale (e.g., Peclet number). Plus, if the authors indicate what method (e.g., upwind difference scheme) they use to determine interfacial properties (e.g., hydraulic conductivity), their work will be better understood by readers.

**Response:**

ELMv0 uses a default tolerance of $10^{-5}$ [kg m$^{-2}$] for water mass balance across each time step (30 minutes). VSFM uses an adaptive timestep to ensure solution of the nonlinear equations over the 30 minute timestep is below that tolerance. The permeability at each control volume interface is obtained by a distance weighted harmonic average as mentioned on line 555-556; while an upwind scheme is used for the term $k_r/\mu$ as given by equation 31. We have clarified these descriptions in the revised manuscript to address the reviewer's comment.

**References**

Clark, M. P., Fan, Y., Lawrence, D. M., Adam, J. C., Bolster, D., Gochis, D. J., Hooper, R. P., Kumar, M., Leung, L. R., Mackay, D. S., Maxwell, R. M., Shen, C., Swenson, S. C., and Zeng, X.: Improving the representation of hydrologic processes in Earth System Models, Water Resources Research, 51, 5929-5956, 2015.

Collier, N., Hoffman, F. M., Lawwrence, D. M., Keppel-Aleks, G., Koven, C. D., Riley, W. J., Mu, M., and Randerson, J. T.: The International Land 1 Model Benchmarking (ILAMB) System: Design, Theory, and Implementation, in review J. Advances in Modeling Earth Systems, 2018. 2018.

Hoffman, F. M., Koven, C. D., Keppel-Aleks, G., Lawrence, D. M., Riley, W. J., Randerson, J. T., Ahlstrom, A., Abramowitz, G., Baldocchi, D. D., Best, M. J., Bond-Lamberty, B., Kauwe}, M. G. D., Denning, A. S., Desai, A. R., Eyring, V., Fisher, J. B., Fisher, R. A., Gleckler, P. J., Huang, M., Hugelius, G., Jain, A. K., Kiang, N. Y., Kim, H., Koster, R. D., Kumar, S. V., Li, H., Luo, Y., Mao, J., McDowell, N. G., Mishra, U., Moorcroft, P. R., Pau, G. S. H., Ricciuto, D. M., Schaefer, K., Schwalm, C. R., Serbin, S. P., Shevliakova, E., Slater, A. G., Tang, J., Williams, M., Xia, J., Xu, C., Joseph, R., and Koch, D.: International Land Model Benchmarking (ILAMB) 2016 Workshop Report, U.S. Department of Energy, Office of Science, 159 pp., 2017.

---

## Author Response (AR1)

 **Lawrence Berkeley National Laboratory**

[Figure]

Earth & Environmental Sciences Area

June 2018

Dear Dr. Kala,

My co-authors and I are pleased to submit revised manuscript entitled "*Development and evaluation of a variably saturated flow model in the global E3SM Land Model (ELM) Version 1.0*" for your consideration as a model description paper in *Geoscientific Model Development*.

We thank the executive editor and the two reviewers for their insightful and constructive feedbacks, which helped us clarify important aspects of our work. Modifications made in the revised version of the manuscript as compared to initial submission are summarized below:

1. As suggested by both reviewers, we have highlight the features of VSFM model and justified the increase in 30% computational cost by updating sections 2.2.1, 3.4, and 3.5.
2. The introduction section has been updated to highlight the need for a unified treatment of soil hydrologic processes in unsaturated and saturated zone, which are met by the VSFM developed in this study.
3. The code availability section has been updated to include details about the model version used in the study and the code is now publicly available at https://github.com/E3SM-Project/E3SM.
4. Additionally, we have added extensive notes on how to reproduce VSFM benchmarking results against observations and PFLOTRAN at https://bitbucket.org/gbisht/notes-for-gmd-2018-44
5. Finally, we have provided detailed response to all comments from the two reviewers.

My co-authors and I believe we have thoroughly addressed all the reviewer comments and that the revised manuscript is well suited for publication in Geoscientific Model Development. We look forward to receiving your response.

Sincerely,
Gautam Bisht

**Lawrence Berkeley National Laboratory**

One Cyclotron Road / MS: 84-332 / Berkeley, California 94720 / **phone** 510-486-5036 / wjriley@lbl.gov

[revised manuscript text omitted]

---

## Referee Report (RR1)

**Referee Report**

**Development and evaluation of a variably saturated flow model in the global E3SM Land Model (ELM) Version 1.0**

In this paper, the authors intended to unify the treatment of subsurface water movement using variably saturated Richards' equation and to demonstrate its performance in predicting groundwater level and other hydrologic components. However, the results do not fully support the motivation shown, and the needed validations are not provided. In order to substantiate the authors' claim, I think some issues below should be addressed.

1. The authors tried to demonstrate the robustness of ELMv1-VSFM by conducting several experimental simulations. However, there may be inaccuracies in numerical solutions due to differences in the size of spatial and temporal mesh (El-Kadi & Ling, 1993). The authors used outputs from simulations with different configurations (e.g. spatial resolution of grid-cell, soil column depth, spatial discretization) to support the robustness of global analysis. Moreover, temporal resolutions used in the experimental simulations are not indicated in the paper. Since information required for ensuring numerical stability of the model is not indicated, I am not sure whether or not ELMv1-VSFM converges well at a spatial-temporal resolution of 1.90 (latitude) × 2.50 (longitude) with a 30 [min] time-step.

2. The authors mentioned that there are advantages to using variably-saturated flow model (variably saturated Richards' equation) rather than applying different governing equations for each flow domain noted in the previous work. However, they did not specify what the relative strengths of using variably saturated Richards' equation are compared to adapting different equations (e.g., computational cost). The reason why they intended to unify the treatment of soil hydrologic processes should be stated.

3.The authors used ILAMB package to show additional consideration of saturated zone does not degrade the model's predictive capabilities in other hydrologic processes. However, without any explanation of the interaction between groundwater and other components (e.g., streamflow, LH/SH), it is difficult to accept the author's claim saying further consideration about groundwater-surface water interaction does not degrade other predictions.

4. The authors mentioned this work has a focus on representing groundwater-surface water interactions but all the outputs appear to be related addressing subsurface hydrology using VSFM. The authors may want to specify what they did to emphasize their focus on groundwater-surface water interactions by adding more results regarding that (e.g., interactive effect between runoff and groundwater level)

I would like to note some recommendations for this paper: 1) how the authors determine the robustness of the model based on the results of the experimental simulations should be specified. 2) the authors may need to perform experimental simulations with the same configuration used for global analysis in order to show the numerical stability of the model. 3) the authors may want to demonstrate the numerical stability of the model by providing some indexes (e.g., Peclet number). 4) the authors can demonstrate the benefits of applying variably-saturated flow model compared to outputs derived from different physics application, especially in terms of computational cost. 5)

the authors may want to add some description about the modeling scheme used for representing the interactions between stream and groundwater and between evapotranspiration and groundwater. 6) To emphasize their motivation for groundwater-surface water interaction, the authors may want to indicate how runoff simulation is correlated groundwater level.

Bibliography

El-Kadi, A. I., & Ling, G. (1993). The Courant and Peclet Number criteria for the numerical
    solution of the Richards Equation. *Water Resources Research*.

---

## Author Response (AR2)

**Lawrence Berkeley National Laboratory**

[Figure]

**Earth & Environmental Sciences Area**

September 2018

Dear Dr. Kala,

My co-authors and I are pleased to submit revised manuscript entitled "*Development and evaluation of a variably saturated flow model in the global E3SM Land Model (ELM) Version 1.0*" for your consideration as a model description paper in *Geoscientific Model Development*.

We very much appreciate the reviewers' comments and feel that they have allowed us to substantially improve our manuscript. Modifications made in the revised version of the manuscript as compared to the last submission are summarized below:

1. We performed additional new 10-years simulations to demonstrate robustness numerical solutions with respect to spatial and temporal resolutions. We have added description of new simulation results to the Supplemental Material, and describe them in the revised Results section.
2. As suggested by reviewer-2, we added additional details on the time integration methodology of the model in the Appendix.
3. Finally, we have provided detailed response to all comments from the two reviewers.

My co-authors and I believe we have thoroughly addressed all the reviewer comments and that the revised manuscript is well suited for publication in Geoscientific Model Development. We look forward to receiving your response.

Sincerely,
Gautam Bisht

**Lawrence Berkeley National Laboratory**

One Cyclotron Road / MS: 84-144 / Berkeley, California 94720 / **phone** 510-486-6246 / gbisht@lbl.gov

**Reviewer #1**

*(1) Eq. (10) - typically Pc is compared against air entry pressure instead of 0.*

**Response:**

Equation 10 has been corrected following the reviewer's suggestion.

*(2) Not all variables are defined in Eq. (10) and (11).*

**Response:**

Definitions of missing variables have been now added in the latest version of the manuscript.

*(3) Variables in Eq. (16) defined in the appendix, but not in the main text.*

**Response:**

Definition of missing variables have been now added in the latest version of the manuscript.

*(4) Redundant "nonlinear" in line 221.*

**Response:**

The text has been updated to remove the redundant 'nonlinear'.

*(5) I don't agree with the statement in lines 233 to 234. At least one can compare water table depth and soil moisture using VSFM and the default scheme within ELM.*

**Response:**

In ELMv0, code for individual process models cannot be built independent of E3SM code base. Thus, individual process models cannot be easily tested against analytical solutions or other model configurations. We have updated the text to clarify this ELMv0 limitation.

*(6) It's not clear which WRC is used for the tests from Table 1 as parameters from both Eqs. 10 and (11) are used. From line 255, Van Genuchten model is used, then n and m are missing in*

**Response:**

All three test problems used the van Genuchten model. The second column of Table 1 originally incorrectly listed values for 'lamba' instead of 'm'. Table 1 has been corrected and the equation for computing 'n' based on 'm' has been added on line 201.

*Table 1.*
*(9) line 298 - what is beta?*

**Response:**

      Beta [radians] is mean grid cell topographic slope and is now described in the updated text.

**Report #2**

*1. The authors tried to demonstrate the robustness of ELMv1-VSFM by conducting several experimental simulations. However, there may be inaccuracies in numerical solutions due to differences in the size of spatial and temporal mesh (El-Kadi & Ling, 1993). The authors used outputs from simulations with different configurations (e.g. spatial resolution of grid-cell, soil column depth, spatial discretization) to support the robustness of global analysis. Moreover, temporal resolutions used in the experimental simulations are not indicated in the paper. Since information required for ensuring numerical stability of the model is not indicated, I am not sure whether or not ELMv1-VSFM converges well at a spatial-temporal resolution of 1.90 (latitude) × 2.50 (longitude) with a 30 [min] time-step.*

**Response:**

      We have now updated Table 1 to include information about the spatial and temporal discretization used for the three single-column benchmark problems. Additionally, Section 3.1 has been updated to include a reference to Table 1. To address the reviewer's comment regarding numerical stability, we performed simulations with higher vertical and temporal resolution, as described below in our responses to Reviewer #2.

*2. The authors mentioned that there are advantages to using variably-saturated flow model (variably saturated Richards' equation) rather than applying different governing equations for each flow domain noted in the previous work. However, they did not specify what the relative strengths of using variably saturated Richards' equation are compared to adapting different equations (e.g., computational cost). The reason why they intended to unify the treatment of soil hydrologic processes should be stated.*

**Response:**

      Clark et al. (2015) summarized opportunities and challenges for improving hydrological processes in the global land surface models and identified that incorporation of variably saturated hydrologic flow models is expected to improve simulation of coupled soil moisture and shallow groundwater dynamics. In the last version of our manuscript, the reference to the Clark et al. (2015) study was included in the Introduction section. In order to state the motivation of our work upfront, we have updated the abstract to include reference to Clark et al. (2015) recommendation for developing a unified treatment of soil hydrologic processes.

*3.The authors used ILAMB package to show additional consideration of saturated zone does not degrade the model's predictive capabilities in other hydrologic processes. However, without any explanation of the interaction between groundwater and other components (e.g., streamflow, LH/SH), it is difficult to accept the author's claim saying further consideration about groundwater- surface water interaction does not degrade other predictions.*

**Response:**

As stated in the revised manuscript starting on line 412, "The International Land Model Benchmarking (ILAMB) package (Hoffman et al., 2017) provides a comprehensive evaluation of predictions of carbon cycle states and fluxes, hydrology, surface energy budgets, and functional relationships by comparison to a wide range of observations". The ILAMB package explicitly compares model predictions of many hydrological and surface energy components, including large river basin flows, LH, SH, etc. In the revised manuscript, Table 3 compares bias, RMSE, and an ILAMB score (which combines metrics associated with spatial and temporal variability, biases, etc.) for LH, SH, TWSA, and large river basin flows.

*4. The authors mentioned this work has a focus on representing groundwater-surface water interactions but all the outputs appear to be related addressing subsurface hydrology using VSFM. The authors may want to specify what they did to emphasize their focus on groundwater-surface water interactions by adding more results regarding that (e.g., interactive effect between runoff and groundwater level)*

**Response:**
      The work we report on here focuses on improving near-surface soil moisture and ground hydrology representation in ELM. We have corrected the abstract in the revised manuscript to clarify this point.

*I would like to note some recommendations for this paper:*
    6) *how the authors determine the robustness of the model based on the results of the experimental simulations should be specified.*

**Response:**
      In order to demonstrate model robustness and flexibility to easily configure the model for a range of problem setups, we performed VSFM simulations for three offline simulation as described in Section 2.3. The problems included infiltration in a dry soil column, infiltration in a layered soil system, and water table dynamics in a variably saturated soil column. For all three problems, VSFM results accurately reproduced published datasets and agreed well with predictions from an existing variably saturated flow model. The benchmark problems used in our study have been previously used to show robustness of variably saturated flow models (Kumar et al., 2009; Shen and Phanikumar, 2010).

*2) the authors may need to perform experimental simulations with the same configuration used for global analysis in order to show the numerical stability of the model.*

**Response:**
Since there is such a large mismatch in temporal (days versus 100's of years) and spatial (2 m versus 150 m deep) scales between the benchmark problems and global simulations, we do not believe it is appropriate to perform 1D benchmark simulations with the spatio-temporal configurations of global simulations. However, we acknowledge that the reviewer has a valid concern about the sensitivity of numerical solutions with respect to spatial and temporal resolutions. Therefore, to address this issue, we performed the following additional new 10-years simulations:

1. SIM_HALF_DT: All configurations were the same as those used in the global simulation with optimal $f_d$ except maximum allowable VSFM timestep was set to 15 min
2. SIM_HALF_DT_AND_HALF_DZ: All configurations were the same as those used in the global simulation with optimal $f_d$ except maximum allowable VSFM timestep was set to 15 min and spatial resolution of ELM was doubled by increasing the number of soil layers to 118 and decreasing the soil thickness for each layer appropriately to keep the total soil column depth fixed at 150 m.

The results are encouraging: the global mean difference in the simulated annual water table depth (WTD) for the 10th year between SIM_OPT and SIM_HALF_DT at 25th, 50th, and 75th percentiles were extremely small (0.001, 0.002 and 0.005 m, respectively). Small difference between SIM_OPT and SIM_HALF_DT_AND_HALF_DZ at 25th, 50th, and 75th percentiles were also found (0.091, 0.488, 0.945 [m], respectively). These results show that simulated WTD is insensitive to VSFM sub timestep, and has small sensitivity to vertical spatial resolution. We have added these simulation results to the Supplemental Material, and describe them in the revised Results section.

*3) the authors may want to demonstrate the numerical stability of the model by providing some indexes (e.g., Peclet number).*

**Response:**
We thank the reviewer for this suggestion, as including more details on numerical properties of the model will improve the manuscript. VSFM uses a two-stage check to determine an acceptable numerical solution:
- Stage-1: At any temporal integration stage, the model attempts to solve the set of nonlinear equations given by Equation 19 with a given timestep. If the model fails to find a solution to the nonlinear equations with a given error tolerance settings, the timestep is reduced by half and the model again attempts to solve the nonlinear problem. If the model fails to find a solution after a maximum number of time step cuts (currently 20), the model reports an error and stops execution. None of the simulations reported in this paper failed this check.
- Stage-2: After a numerical solution for the nonlinear problem is obtained, a mass balance error is calculated as the difference between input and output fluxes and change in mass over the integration timestep. If the mass balance error exceeds $10^{-5}$ kg m$^{-2}$, the error tolerances for the nonlinear problem are tightened by a factor of 10 and the model re-enters Stage-1. If the model fails to find a solution with an acceptable mass balance error after 10 attempts of tightening error tolerances, the model reports an error and stops execution. None of the simulations reported in this paper failed this check.

We extended the Appendix to include a new section, 5.4, to details about the time integration methodology of VSFM, as described above.

*4) the authors can demonstrate the benefits of applying variably-saturated flow model compared to outputs derived from different physics application, especially in terms of computational cost.*

**Response:**

As described in Section 3.3 of the manuscript, we performed computational cost calculations for VSFM using 96, 192, 384, 768, and 1536 cores. Compared to the default hydrological model, VSFM is ~30% more expensive on an optimal processor layout. To address this reviewer comment, we have additionally added a figure showing the performance of the default model and VSFM at different core counts in Supplementary material.

*5) the authors may want to add some description about the modeling scheme used for representing the interactions between stream and groundwater and between evapotranspiration and groundwater.*

**Response:**

VSFM does not explicitly represent stream and groundwater interactions. VSFM is driven by a vertically prescribed source/sink of water over the soil column, which has been calculated by other components of ELM (e.g., transpiration, infiltration). Section 2.2 has been updated to describe how all sources and sinks of water are handled in VSFM.

*6) To emphasize their motivation for groundwater-surface water interaction, the authors may want to indicate how runoff simulation is correlated groundwater level.*

**Response:**

As indicated in Table 3, runoff (which includes subsurface and surface components) does depend on groundwater depth, and globally the change in the ILAMB score using default and optimal drainage parameter is only 0.02 m. As mentioned above in response to reviewer #1's comment, the focus of this work is on near-surface soil moisture and GW, and not on effects to surface water dynamics. We have clarified this point in the Abstract and Introduction of the revised manuscript.

[revised manuscript text omitted]